# Dual Variance Reduction with Momentum for Imbalanced Black-Box Discrete Prompt Learning

## Abstract

Black-box prompt learning has proven to be an effective approach for customizing large language models (LLMs) offered as services to address various downstream tasks. Within this domain, policy gradient-based methods have garnered substantial attention as a prominent approach for learning discrete prompts. However, the highly imbalanced data distribution in the real world limits the applicability of such approaches by influencing LLMs' tendency to favor certain categories. To tackle the challenge posed by imbalanced data, this paper pioneers the integration of pairwise AUC loss into the policy gradient optimization of discrete text prompts and proposes learning discrete prompts with doubly policy gradient. Unfortunately, the doubly policy gradient estimation suffers from two variance components, resulting in unstable optimization. As a further improvement, we propose (1) a novel unbiased variance-reduced doubly policy gradient estimator and (2) incorporating the STORM variance reduction technique. Ultimately, we introduce a novel ***momentum-based discrete prompt learning method with doubly policy gradient*** (mDP-DPG). Crucially, we provide theoretical convergence guarantees for mDP-DPG within standard frameworks. The experimental results show that mDP-DPG surpasses baseline approaches across diverse imbalanced text classification datasets, emphasizing the advantages of our proposed approach for tackling data imbalance. Our code is available at the following URL: https://anonymous.4open.science/r/DPDPG-1ECB.

## 1 Introduction

Large language models (LLMs) have achieved milestone accomplishments on a wide range of natural language processing (NLP) tasks (Brown et al., 2020; Devlin et al., 2018; Raffel et al., 2020). However, the increasing parameters pose challenges for tuning. Prompting has emerged as the parameter-efficient paradigm for adapting LLMs to specific NLP tasks (Li & Liang, 2021; Lester et al., 2021; Liu et al., 2023). Well-crafted prompts can effectively enhance the performance of LLMs on various downstream tasks instead of retraining. Recently, considering that most existing LLMs, such as GPT-4, only provide cloud-based API services, researchers have introduced the Language-Model-as-a-Service scenario, where users are limited to interacting with LLMs solely through APIs, creating a black-box setting (Sun et al., 2022b;a). Black-box prompt learning has been an effective strategy for adapting LLMs to downstream tasks due to the opacity of black-box LLMs (Deng et al., 2022; Prasad et al., 2022; Diao et al., 2022).

Currently, much attention has been focused on black-box discrete prompt learning, with policy gradient-based methods becoming highly influential (Diao et al., 2022; Lin et al., 2023). Within these investigations, discrete prompt learning is treated as a distribution optimization problem, using policy gradient to update the prompt distribution and overcome the issue of inaccessible gradients in black-box LLMs. However, these efforts focus solely on vanilla text classification without any additional handling of imbalanced data. These imply that adapting them to address the class imbalance problem to bridge the gap between LLM services and downstream tasks while providing theoretical convergence guarantees remains a significant challenge.

Departing from idealized situations, real-world data usually features severe class distribution imbalances, where certain minority classes are markedly less prevalent than majority classes in classification problems (Henning et al., 2022). For example, non-hateful tweets are more prevalent than hateful ones on Twitter (Waseem & Hovy, 2016), and the positive and negative reviews on Amazon are also imbalanced. Specifically, the ratio of negative to positive movie reviews is approximately 6.24, while it is as high as 9.75 for book reviews (Gao et al., 2021).

Simultaneously, imbalanced class distributions hinder prompt learning by making LLMs more likely to prioritize well-represented classes, which in turn lowers prompt performance. Dong et al. (2022) initially identified this phenomenon within the computer vision (CV) domain, noting that prompts could benefit models on long-tail data, but their performance still lagged far behind the state-of-the-art. Similarly, class imbalance issues also impact the NLP black-box prompt learning. When the downstream task is a text classification problem, cross-entropy is typically chosen to build the objective function. However, this becomes unreasonable in the presence of imbalanced data since the minority class data have little influence, leading to the aforementioned issues (Liu et al., 2020).

A common approach in previous research for addressing imbalanced class distributions is to use AUC (Area Under the ROC Curve) maximization as the optimization objective. The AUC is defined as the probability that the prediction score for a positive example exceeds that for a negative example in statistical terms (Hanley & McNeil, 1982; 1983), and AUC maximization is proposed to address the imbalanced data problem (Zhao et al., 2011; Yuan et al., 2020). However, high variance emerges as a major challenge in the learning process of prompts in two respects. Firstly, the sampling of prompts from the prompt distribution introduces inherent randomness, making the optimization process unstable. Secondly, sampling both positive and negative examples for AUC maximization will also introduce high variance. In scenarios with imbalanced classes, the sampling of data for training not only amplifies the disparity between majority and minority classes but also increases the variance of gradient estimation during optimization. This duality of variance poses significant difficulties in effectively learning prompts that generalize well across all classes, particularly in settings with highly imbalanced data.

In order to tackle the above problems, we propose a novel ***momentum-based discrete prompt learning method with doubly policy gradient*** (mDP-DPG) that utilizes AUC maximization to adapt LLMs to downstream tasks with imbalanced data. By minimizing AUC's pairwise surrogate loss using policy gradient, mDP-DPG prevents prompts from being degraded by the majority class, thereby preserving performance. Moreover, due to the doubly sampling of examples during gradient estimation, we refer to the policy gradient in mDP-DPG as the doubly sampled policy gradient, abbreviated as doubly policy gradient. We further propose an unbiased, variance-reduced doubly policy gradient estimator (VR-DPGE) to improve convergence in practice. While the estimator suffers from variance, stochastic sampling of mini-batches from the dataset also introduces variance into gradient estimation. To reduce the dual variance, we introduce the momentum-based variance reduction strategy STORM (Cutkosky & Orabona, 2019; Huang et al., 2020). STORM does not rely on constructing variance-reduced gradients through giant batch sizes, as SVRG does (Johnson & Zhang, 2013; Xiao & Zhang, 2014). Instead, it employs a variant of momentum, making it can be seamlessly integrated into the optimization of pairwise AUC loss for variance reduction. Additionally, unlike other policy gradient-based methods for black-box discrete prompt learning, mDP-DPG has rigorous theoretical convergence guarantees.

The main contributions of this work are summarized as follows:

1. We propose a novel momentum-based discrete prompt learning method named mDP-DPG, which introduces VR-DPGE and STROM strategy to address challenges posed by dual variance. Using pairwise AUC loss as objective function, mDP-DPG preserves prompts' performance in downstream tasks with class imbalance. As far as we know, we are the first to discuss how to address imbalanced data in, NLP prompt learning.

2. We establish rigorous theoretical analysis of the mDP-DPG. Specifically, we provide proof that mDP-DPG achieves an oracle complexity of $O(1/\epsilon^3)$, validating its effectiveness in optimizing the pairwise AUC loss for black-box prompt learning in the context of imbalanced data.

3. Numerous experimental results on RoBERTa-large, GPT2-XL, and Llmma3 show that our method achieve state-of-the-art across various class imbalance datasets, demonstrating the

effectiveness of prompts learned through mDP-DPG on imbalanced data. Furthermore, our research findings confirm that imbalanced data negatively impacts prompt learning, emphasizing the importance of imbalanced prompt learning.

## 2 RELATED WORKS

**Black-Box Prompt Learning.** There is a significant amount of research focusing on black-box prompt learning, which has achieved promising results in NLP tasks (Brown et al., 2020; Prasad et al., 2022; Han et al., 2023; Hou et al., 2023). Prompts come in two formats: continuous and discrete. The continuous prompt is a series of vectors, which concatenates with token embeddings. Sun et al. (2022b) propose the black-box tuning (BBT) framework, which optimizes prompts in low-dimensional subspace and obtains continuous prompts in the original space through a random matrix. Sun et al. (2022a) present an improved version of BBT that adds continuous prompt prefixes to each hidden layer of LLM. Zheng et al. (2023) point out the inappropriateness of random matric in Sun et al. (2022b) and leverage meta-learning to identify the optimal subspace. On the other hand, the discrete prompt is a sequence of tokens, which is more appropriate for real applications. Deng et al. (2022) formulate discrete prompt learning as a reinforcement learning problem and generate discrete prompts using policy network. Diao et al. (2022) utilize policy gradients to optimize the distribution of the discrete prompt. Zhao et al. (2023) introduce a genetic algorithm to search for discrete prompts guided by LLMs predictive probabilities.

**AUC Maximization.** AUC maximization is a machine learning paradigm aimed at maximizing the AUC score of models. A substantial amount of research has been dedicated to this topic, integrating it with various contexts such as supervised learning (Joachims, 2005), semi-supervised learning (Iwata et al., 2020), online learning (Gao et al., 2016), and federated learning (Yuan et al., 2021). Many algorithms frequently minimize the pairwise surrogate loss for AUC maximization. Zhao et al. (2011) propose two online AUC maximization algorithms, which utilize the reservoir sampling technique to avoid memorizing all training samples. Gao et al. (2016) focus on the challenge of optimizing with only a single pass of training samples and propose a regression-based algorithm using square surrogate loss. The issue of the pairwise surrogate loss lies in the necessity to construct sample pairs from opposite classes. To overcome this challenge, Ying et al. (2016) demonstrate the equivalence between AUC optimization and the saddle point problem. Liu et al. (2020) extend the aforementioned equivalence to the case of deep neural network models.

**Variance Reduction.** In optimization problems, variance reduction is a frequently utilized improvement method (Cutkosky & Orabona, 2019). Numerous variance reduction techniques necessitate setting gradient checkpoints and calculating gradients at these checkpoints with giant batch sizes Johnson & Zhang (2013); Fang et al. (2018); Zhou et al. (2018). Cutkosky & Orabona (2019) propose a variance reduction technique based on the momentum variant, termed STORM, which facilitates variance reduction in non-convex optimization without giant batch sizes. Huang et al. (2020) propose a similar variance reduction technique for zero-order optimization to accelerate black-box minimization and minimax optimization problems.

## 3 METHODOLOGY

In this section, we first present the problem formulation in Section 3.1, where we define discrete prompt learning with the pairwise AUC loss as the objective function. Subsequently, in Sections 3.2 and 3.3, we discuss how to address the challenges posed by the dual variance in optimization. Specifically, in Section 3.2, we present VR-DPGE to reduce the variance introduced by prompt sampling in the doubly policy gradient estimation. In Section 3.3, we introduce a momentum-based variance reduction technique to reduce the dual variance.

**Notations.** Let $\mathcal{D} \triangleq \{(\boldsymbol{X}_1, y_1), (\boldsymbol{X}_2, y_2), \ldots, (\boldsymbol{X}_M, y_M)\}$ denote a set of training data with cardinality $M$. For any $m \in \{1, 2, \ldots, M\}$, $\boldsymbol{X}_m$ represents an input training example (e.g., a piece of text), and $y_m \in \{-1, 1\}$ denotes its corresponding label. We use $T(\cdot)$ to represent a tokenizer that converts an input text to a token vector, and $\mathbf{S}_m \triangleq T(\boldsymbol{X}_m)$ as the $m$-th token vector. Let $\tilde{\mathcal{D}} \triangleq \{(\mathbf{S}_1, y_1), (\mathbf{S}_2, y_2), \ldots, (\mathbf{S}_M, y_M)\}$ be the set of tuples composed of token vectors and labels. Discrete prompt is defined as a token vector with $n$ discrete tokens $\mathbf{T} = [t_1, t_2, \ldots, t_n]$. For any

i ∈ {1, 2, . . . , n}, the word $t_i \in \mathcal{W}$ and $\mathcal{W}$ is the set consisting of tokens from a given vocabulary. Let $\mathbf{S} \in \{\mathbf{S}_1, \mathbf{S}_2, \dots, \mathbf{S}_M\}$ denotes any token vectors, then the user query to a black-box LLM $h(\cdot)$ is denoted as $h([\mathbf{T}, \mathbf{S}])$, i.e. $h([\mathbf{T}, \mathbf{S}])$ denote the prediction of the LLM $h(\cdot)$ on an input $[\mathbf{T}, \mathbf{S}]$. For $p \in \mathbb{N}^*$, $\mathbf{1}_p$ denotes the vector of size $p$ composed only of ones.

## 3.1 PROBLEM STATEMENT

**Discrete Prompt Learning.** Discrete prompt learning aims to learn a discrete textual prompt consisting of $n$ tokens, which is a more pragmatic scenario. Diao et al. (2022) formulate discrete prompt learning as a distribution optimization problem. Specifically, they assume each token of the prompt is sampled from the independent categorical distribution $t_i \sim \text{Cat}(\boldsymbol{p}_i)$, where $\boldsymbol{p}_i \in \mathcal{C}$ and $\mathcal{C} = \{\boldsymbol{x} : \|\boldsymbol{x}\|_1 = 1, 0 \le \boldsymbol{x} \le 1\}$. The component $\boldsymbol{p}_{i,j}$ denotes the probability of sampling the $j$-th token from the vocabulary $\mathcal{W}$. By denoting $\boldsymbol{\mathcal{C}}$ the subset of $\mathbb{R}^{|\mathcal{W}| \times n}$ such that for any $\boldsymbol{p} \in \boldsymbol{\mathcal{C}}$ (where $\boldsymbol{p}_i$ denotes a column of $\boldsymbol{p}$), $\boldsymbol{p}_i \in \mathcal{C}$, the objective function during optimization is as follows:

$$\min_{\boldsymbol{p} \in \boldsymbol{\mathcal{C}}} \mathbb{E}_{(\mathbf{S}, y)} \mathbb{E}_{\mathbf{T}}[\ell(h([\mathbf{T}, \mathbf{S}]), y)] = \min_{\boldsymbol{p} \in \boldsymbol{\mathcal{C}}} \mathbb{E}_{(\mathbf{S}, y)} \Sigma_{\mathbf{T}} \ell(h([\mathbf{T}, \mathbf{S}]), y) P(\mathbf{T}) \tag{1}$$

where $\ell(\cdot)$ is the loss function that evaluates the prediction of the black-box LLM $h(\cdot)$ based on the ground truth label $y$. $P(\mathbf{T}) = \prod_{i=1}^{n} P(t_i)$ is the joint probability of the discrete prompt $\mathbf{T}$. To avoid confusion, we refer to $\boldsymbol{p}_i$ as the token distribution and the joint distribution $\boldsymbol{p}$ of $n$ token distributions as the prompt distribution.

**AUC maximization.** AUC is a common metric for evaluating model performance in imbalanced binary classification problems and AUC maximization is a form of pairwise learning that aims to maximize the AUC score during the model training. By employing the square loss as the surrogate, which is statistically consistent with AUC (Gao & Zhou, 2012), AUC maximization can be formulated as

$$\arg \min_{f} \mathbb{E}[(1 - f(x) + f(x'))^2 | y = +1, y' = -1] \tag{2}$$

**Discrete Prompt Learning with AUC maximization.** We propose employing pairwise AUC loss for black-box discrete prompt learning to address the challenge posed by class imbalance in prompt learning. Therefore, we can formulate the objective of black-box prompt learning that minimizes the expected risk $\mathcal{L}(\boldsymbol{p})$ as follows

$$\boldsymbol{p}^* = \arg \min_{\boldsymbol{p} \in \boldsymbol{\mathcal{C}}} \mathcal{L}(\boldsymbol{p}) = \arg \min_{\boldsymbol{p} \in \boldsymbol{\mathcal{C}}} \mathbb{E}_{(\mathbf{S}, y)} \mathbb{E}_{(\mathbf{S}', y')} \mathbb{E}_{\mathbf{T}} L(h([\mathbf{T}, \cdot]), (\mathbf{S}, y), (\mathbf{S}', y')) \tag{3}$$

where $L(h([\mathbf{T}, \cdot]), (\mathbf{S}, y), (\mathbf{S}', y'))$ is the pairwise AUC square loss given a discrete prompt $\mathbf{T}$ and a pair of samples $(\mathbf{S}, y), (\mathbf{S}', y')$.

$$L(h([\mathbf{T}, \cdot]), (\mathbf{S}, y), (\mathbf{S}', y')) = \begin{cases} (1 - h([\mathbf{T}, \mathbf{S}]) + h([\mathbf{T}, \mathbf{S}']))^2, \text{if } y = +1 \text{ and } y' = -1, \\ 0, \text{otherwise.} \end{cases} \tag{4}$$

In real-world applications, instead of minimizing the expected risk $\mathcal{L}(\boldsymbol{p})$, we consider an independent and identically distributed training set $\tilde{\mathcal{D}}$ and the empirical risk $\mathcal{L}_M(\boldsymbol{p})$ of the pairwise loss function on $\tilde{\mathcal{D}}$ is as follows

$$\boldsymbol{p}^* = \arg \min_{\boldsymbol{p} \in \boldsymbol{\mathcal{C}}} \mathcal{L}_M(\boldsymbol{p}) = \arg \min_{\boldsymbol{p} \in \boldsymbol{\mathcal{C}}} \frac{1}{M(M-1)} \Sigma_{i,j \in \tilde{\mathcal{D}}, i \neq j} \underbrace{\mathbb{E}_{\mathbf{T}} L(h([\mathbf{T}, \cdot]), (\mathbf{S}_i, y_i), (\mathbf{S}_j, y_j))}_{F_{i,j}(\boldsymbol{p})} \tag{5}$$

## 3.2 REDUCE VARIANCE IN PROMPT SAMPLING WITH VR-DPGE

Now black-box discrete prompt learning with AUC maximization transforms into the task of solving the black-box pairwise learning problem. Given a pair of samples $(\mathbf{S}_i, y_i)$ and $(\mathbf{S}_j, y_j)$, we can formulate doubly stochastic gradient $\nabla_{\boldsymbol{p}} F_{i,j}(\boldsymbol{p})$ as equation 6 with the aid of the policy gradient estimator for solving problem 5. Due to the double sampling, we refer to equation 6 doubly sampled policy gradient, abbreviated as doubly policy gradient.

$$\begin{aligned} \nabla_{\boldsymbol{p}} F_{i,j}(\boldsymbol{p}) &= \nabla_{\boldsymbol{p}} \mathbb{E}_{\mathbf{T}} L(h([\mathbf{T}, \cdot]), (\mathbf{S}_i, y_i), (\mathbf{S}_j, y_j)) \\ &= \Sigma_{\mathbf{T}} L(h([\mathbf{T}, \cdot]), (\mathbf{S}_i, y_i), (\mathbf{S}_j, y_j)) \nabla_{\boldsymbol{p}} P(\mathbf{T}) \\ &= \Sigma_{\mathbf{T}} L(h([\mathbf{T}, \cdot]), (\mathbf{S}_i, y_i), (\mathbf{S}_j, y_j)) P(\mathbf{T}) \nabla_{\boldsymbol{p}} \log P(\mathbf{T}) \\ &= \mathbb{E}_{\mathbf{T}} L(h([\mathbf{T}, \cdot]), (\mathbf{S}_i, y_i), (\mathbf{S}_j, y_j)) \nabla_{\boldsymbol{p}} \log P(\mathbf{T}) \end{aligned} \tag{6}$$

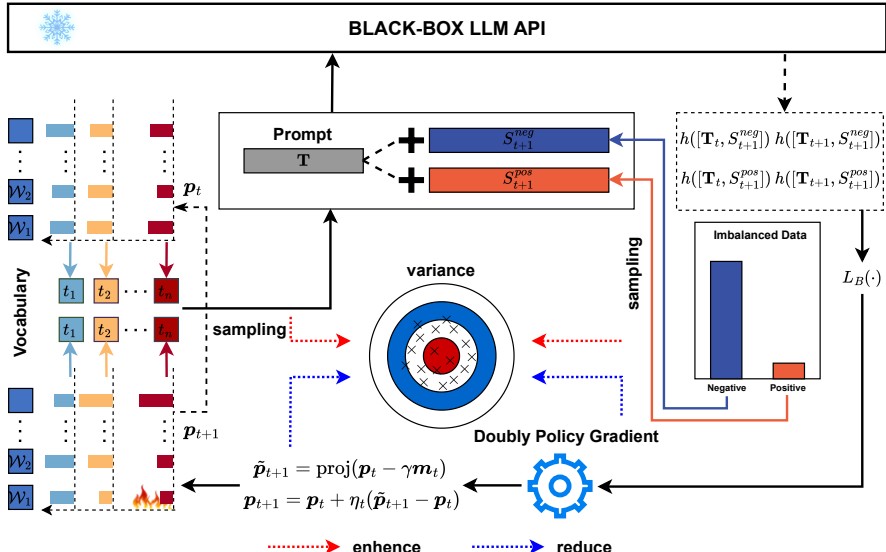

Figure 1: Overview of mDP-DPG. In each iteration, the positive and negative example batches are concatenated with the sampled prompts and input into the LLM to obtain predictions. $S_{t+1}^{pos}$ and $S_{t+1}^{neg}$ represent the sets of sampled positive and negative examples in the $t$-th iteration, respectively. $\mathbf{T}_t$ are prompts sampled from distribution $\boldsymbol{p}_t$

where $P(\mathbf{T}) = \prod_{i=1}^{n} P(t_i)$ denotes the joint probability of the prompt $\mathbf{T}$. Considering $t_i = \mathcal{W}[j_i]$, i.e. the $i$-th token in $\mathbf{T}$ is the $j_i$-th token in $\mathcal{W}$ [1], and $t_i \sim \text{Cat}(\boldsymbol{p}_i)$, we can give explicitly $\nabla_{\boldsymbol{p}} \log P(\mathbf{T})$ as follow (proof in Appendix A),

$$\nabla_{\boldsymbol{p}_{i,j}} \log P(t_i) = \begin{cases} \frac{1}{\boldsymbol{p}_{i,j_i}} & j = j_i \\ 0 & j \neq j_i \end{cases} \tag{7}$$

However, due to the randomness introduced by prompt sampling, the doubly policy gradient suffers from high variance, which adversely affects convergence, just as in policy gradient estimation (Sutton et al., 1999; Rezende et al., 2014). We implement VR-DPGE, which incorporates a baseline subtraction term to reduce variance (Greensmith et al., 2004). Compared to the high variance of loss values, the difference between the loss value and the baseline term has a lower variance, which facilitates more stable gradient estimation. By defining mini-batch $S = \{(\mathbf{S}_q^{pos}, y_q^{pos}), (\mathbf{S}_q^{neg}, y_q^{neg})\}_{q=1}^{B}$ and $L_B(h([\mathbf{T}, \cdot]), S) = \frac{1}{B}\Sigma_q L(h([\mathbf{T}, \cdot]), (\mathbf{S}_q^{pos}, y_q^{pos}), (\mathbf{S}_q^{neg}, y_q^{neg}))$, we can replace $\nabla_{\boldsymbol{p}} F_{i,j}(\boldsymbol{p})$ by VR-DPGE $\boldsymbol{g_p}$ as equation 8.

$$\boldsymbol{g_p} := L_{avg} \mathbf{1}_{|\mathcal{W}|} \mathbf{1}_n^{\top} + \frac{1}{I-1}\Sigma_{\mathbf{k}} \nabla_{\boldsymbol{p}} \log P(\mathbf{T}^{(\mathbf{k})})(L_B(h([\mathbf{T}^{(\mathbf{k})}, \cdot]), S) - L_{avg}) \tag{8}$$

with $L_{avg} := \frac{1}{I}\Sigma_{\mathbf{k}} L_B(h([\mathbf{T}^{(\mathbf{k})}, \cdot]), S)$. $\mathbf{T}^{(\mathbf{k})}$ represents the $k$-th prompt obtained from $I$ prompt samplings. $L_{avg}\mathbf{1}_{|\mathcal{W}|}\mathbf{1}_n^{\top}$ is the unbiased correction term to ensure unbiasedness of the VR-DPGE and the theoretical guarantee is Lemma 1.

### 3.3 REDUCE DUAL VARIANCE WITH MOMENTUM TECHNIQUE

The sampling of positive and negative examples also introduces high variance, especially in the case of highly imbalanced data, and the dual variance from this and prompt sampling poses substantial difficulties for imbalanced discrete prompt learning. However, various variance reduction techniques typically require giant batch sizes to compute the gradient in the checkpoint, such as SVRG, which is difficult to apply to pairwise learning because the number of positive and negative sample

---

[1] The vocabulary $\mathcal{W}$ is constructed following Diao et al. (2022)

combinations is substantial. Therefore, we further employ the momentum-based variance reduction strategy STORM. Specifically, mDP-DPG uses a variant of momentum $\boldsymbol{m}_{t+1}$ (line 16 in Algorithm 1) as the update direction. The content of Lemma 4 demonstrates that the momentum-based strategy effectively reduces the dual variance. The bound in Lemma 4 contains terms that decay with the momentum parameter $\theta_t$, showing that variance of the moving average $\boldsymbol{m}_{t+1}$ is systematically reduced over iterations, leading to a more stable and accurate approximation of the true gradient.

As illustrated in Algorithm 1 and Figure 1, mDP-DPG addresses the class imbalance issue in downstream tasks by minimizing the pairwise AUC surrogate loss. In each iteration, we randomly pick $B$ positive and negative sample pairs from $\tilde{\mathcal{D}}$ to compute pairwise loss. To estimate the doubly policy gradient, we sample $I$ prompts according to the prompt distribution $\boldsymbol{p}_{t+1}$ in the current iteration. Specifically, $\boldsymbol{p}_{t+1,i}$ is the $i$-th token distribution of the prompt distribution $\boldsymbol{p}_{t+1}$ updated by momentum technique in the $t$-th iteration. For the sampled prompt $\mathbf{T}^{(\mathbf{k})}$, we concatenate it with all positive and negative samples in the mini-batch to construct the input for the LLM. Then, we calculate $\boldsymbol{g}_{\boldsymbol{p}_{t+1},S_t}$ based on LLM predictions and similarly obtain $\boldsymbol{g}_{\boldsymbol{p}_t,S_t}$. Ultimately, we can determine the update direction $\boldsymbol{m}_{t+1}$ for the next iteration and $\text{proj}_{\boldsymbol{\mathcal{C}}}(\cdot)$ in the update step is a projection function that projects updated prompt distribution to the constraint set $\boldsymbol{\mathcal{C}}$ following Diao et al. (2022).

---

**Algorithm 1** mDP-DPG

---

**Input:** Dataset $\tilde{\mathcal{D}}$, hyperparameters $k, \xi, c, \gamma$
**Initialization:** Construct $S_1 = \{(\mathbf{S}_q^{pos}, y_q^{pos}), (\mathbf{S}_q^{neg}, y_q^{neg})\}_{q=1}^B$ from $\tilde{\mathcal{D}}$ in the same way as Line 5-9, then compute $\boldsymbol{m}_1 = \boldsymbol{g}_{\boldsymbol{p}_1,S_1}$.
1: **for** $t = 1, \ldots, T$ **do**
2:      Learning rate $\eta_t = \frac{k}{(\xi+t)^{1/3}}$
3:      Update $\tilde{\boldsymbol{p}}_{t+1} = \text{proj}_{\boldsymbol{\mathcal{C}}}(\boldsymbol{p}_t - \gamma \boldsymbol{m}_t)$, $\boldsymbol{p}_{t+1} = \boldsymbol{p}_t + \eta_t(\tilde{\boldsymbol{p}}_{t+1} - \boldsymbol{p}_t)$
4:      Compute $\theta_{t+1} = c\eta_t^2$
5:      $S_{t+1} = \varnothing$
6:      **for** $q \leq B$ **do**
7:          Sample positive $(\mathbf{S}_q^{pos}, y_q^{pos})$ and negative $(\mathbf{S}_q^{neg}, y_q^{neg})$ examples from $\tilde{\mathcal{D}}$, respectively.
8:          $S_{t+1} = S_{t+1} \cup \{(\mathbf{S}_q^{pos}, y_q^{pos}), (\mathbf{S}_q^{neg}, y_q^{neg})\}$
9:      **end for**
10:     **for** $\text{k} \leq I$ **do**
11:        Sample $j_1^{(\mathbf{k})} \sim \text{Cat}(\boldsymbol{p}_{t+1,1}), \ldots, j_n^{(\mathbf{k})} \sim \text{Cat}(\boldsymbol{p}_{t+1,n})$
12:        $\mathbf{T}^{(\mathbf{k})} = t_1^{(\mathbf{k})} \ldots t_n^{(\mathbf{k})} = \mathcal{W}[j_1^{(\mathbf{k})}] \ldots \mathcal{W}[j_n^{(\mathbf{k})}]$
13:     **end for**
14:     $L_{avg} = \frac{1}{I}\Sigma_{\mathbf{k}}L_B(h([\mathbf{T}^{(\mathbf{k})}, \cdot]), S_{t+1})$
15:     $\boldsymbol{g}_{\boldsymbol{p}_{t+1},S_{t+1}} = L_{avg}\mathbf{1}_{|\mathcal{W}|}\mathbf{1}_n^\top + \frac{1}{I-1}\Sigma_{\mathbf{k}}\nabla_{\boldsymbol{p}_{t+1}}\log P(\mathbf{T}^{(\mathbf{k})})(L_B(h([\mathbf{T}^{(\mathbf{k})}, \cdot]), S_{t+1}) - L_{avg})$
16:     Compute $\boldsymbol{m}_{t+1} = \boldsymbol{g}_{\boldsymbol{p}_{t+1},S_{t+1}} + (1 - \theta_{t+1})[\boldsymbol{m}_t - \boldsymbol{g}_{\boldsymbol{p}_t,S_{t+1}}]$
17: **end for**
**Output:** $\boldsymbol{p}_R$ with $R$ chosen uniformly at random in $\{1, \ldots, T\}$. ($\boldsymbol{p}_T$ in practice).

---

## 4   CONVERGENCE ANALYSIS

In this section, we provide theoretical convergence guarantees for the proposed mDP-DPG algorithm. We first introduce some necessary assumptions and definitions. Then, we analyze the convergence properties of mDP-DPG, showing that it achieves an oracle complexity of $O(1/\epsilon^3)$.

**Lemma 1** (Unbiasedness of the VR-DPGE estimator, Proof in Appendix B.1). *Consider $\boldsymbol{g}_{\boldsymbol{p}}$ the VR-DPGE policy gradient estimator in equation 8. For any $\boldsymbol{p}$ in the constraint set (i.e. such that each $\boldsymbol{p}_i$ for $i$ in $[n]$ defines a categorical probability distribution), such an estimate is an unbiased estimate of the policy gradient, i.e.:*

$$\mathbb{E}_S\mathbb{E}_{\{\mathbf{T}^{(\mathbf{k})}\}_{k=1}^I}\boldsymbol{g}_{\boldsymbol{p}} = \nabla_{\boldsymbol{p}}\mathbb{E}_S\mathbb{E}_{\mathbf{T}}L_B(h([\mathbf{T}, \cdot]), S).$$

**Assumption 1** (Finiteness of the loss). *We assume that there is a constant $C > 0$ such that for any prompt $\mathbf{T}$ and any batch of data $S$, we have $|L_B(h([\mathbf{T}, \cdot]), S)| \leq C$.*

**Remark 1.** *Note that if one uses a loss $L_B(h([\mathbf{T}, \cdot]), S)$ which could be arbitrary large (for instance if $L_B$ is the cross-entropy function), in practice one can always clip such value to ensure boundedness (indeed, since we consider black-box optimization through reinforcement learning in our paper, even if the clipping operation is non-differentiable, optimization of such loss function will still be possible).*

**Lemma 2** (Smoothness of the loss, Proof in Appendix B.2). *Let us denote $\mathcal{C}$ the subset of $\mathbb{R}^{|\mathcal{W}| \times n}$ such that for any $\boldsymbol{p} \in \mathcal{C}$, $\boldsymbol{p}_i \in \mathcal{C}$ (where $\boldsymbol{p}_i$ denotes a column of $\boldsymbol{p}$). Let us denote $\mathcal{P}_{\boldsymbol{p}}$ the probability distribution on prompts $\mathbf{T}$ parameterized by $\boldsymbol{p} \in \mathcal{C}$. Then, $\mathbb{E}_S \mathbb{E}_{\mathbf{T} \sim \mathcal{P}_{\boldsymbol{p}}} L_B(h([\mathbf{T}, \cdot]), S)$ is a smooth-function of $\boldsymbol{p}$ on its domain, with constant $L = \sqrt{n|\mathcal{W}|}C$, that is, for any $(\boldsymbol{p}, \boldsymbol{p}') \in \mathcal{C}^2$:*

$$\|\nabla \mathbb{E}_S \mathbb{E}_{\mathbf{T} \sim \mathcal{P}_{\boldsymbol{p}}} L_B(h([\mathbf{T}, \cdot]), S) - \nabla \mathbb{E}_S \mathbb{E}_{\mathbf{T} \sim \mathcal{P}_{\boldsymbol{p}'}} L_B(h([\mathbf{T}, \cdot]), S)\| \leq L\|\boldsymbol{p} - \boldsymbol{p}'\|. \tag{9}$$

**Assumption 2** (Boundedness of the variance of the gradient). *We assume the following bound on the variance of the VR-DPGE gradient estimator. For any $\boldsymbol{p} \in \mathcal{C}$:*

$$\mathbb{E}_S \mathbb{E}_{\mathbf{T}} \left\| \boldsymbol{g}_{\boldsymbol{p}} - \nabla_{\boldsymbol{p}} \mathcal{L}(\boldsymbol{p}) \right\|^2 \leq \sigma_1^2 / I + \sigma_2^2 / B,$$

*where $\sigma_1$ denotes the variance introduced by the random selection of vocabularies and $\sigma_2$ denotes the variance introduced by the random selection of pairs of positive-negative examples.*

**Remark 2** (Proof in Appendix B.3). *Even if Assumption 2 is not verified for $\mathcal{C} = \{\boldsymbol{p} \in \mathbb{R}^{|\mathcal{W}| \times n} : \forall i \in [n], \|\boldsymbol{p}_i\|_1 = 1, \forall j \in [|\mathcal{W}|], 0 \leq \boldsymbol{p}_{j,i} \leq 1\}$, it is actually verified if one ensures some lower bound on the values of $\boldsymbol{p}$, i.e. it is verified on the set $\mathcal{C} = \{\boldsymbol{p} \in \mathbb{R}^{|\mathcal{W}| \times n} : \forall i \in [n], \|\boldsymbol{p}_i\|_1 = 1, \forall j \in [|\mathcal{W}|], \nu \leq \boldsymbol{p}_{j,i} \leq 1\}$, for some $\nu \in (0, 1]$, as we prove in Appendix B.3. In the experiments however, we could take $\nu = 0$ (i.e. we could keep the original constraint $\mathcal{C}$), which still worked well in practice.*

## 4.1 Convergence Results

Convergence for projected stochastic gradient descent is usually measured in terms of the expected squared norm of the gradient mapping, which we will define below. Since we proved above that the function we consider is smooth and the stochastic gradient is bounded, and the set we project onto is bounded, we can establish the following convergence result:

**Theorem 1** (Convergence rate of mDP-DPG, proof in Appendix B.4). *Suppose that $\{\boldsymbol{p}_t\}_{t=1}^T$ are generated from mDP-DPG. Let $\eta_t = \frac{k}{(\xi+t)^{1/3}}$, $0 < \gamma \leq \min\{\frac{\xi^{1/3}}{2Lk}, \frac{1}{2\sqrt{2}L}\}$, $c \geq \frac{2}{3k^3} + \frac{5}{4}$, $\xi \geq \max\{2, k^3, c^3 k^3\}$, then we have*

$$\frac{1}{T} \sum_{t=1}^{T} \mathbb{E}\|G_{\mathcal{C}}(\boldsymbol{p}_t, \gamma)\| \leq \frac{2\sqrt{2M}}{T^{1/2}} \xi^{1/6} + \frac{2\sqrt{2M}}{T^{1/3}}, \tag{10}$$

*where $G_{\mathcal{C}}(\boldsymbol{p}_t, \eta_t) := \frac{1}{\eta_t} \left(\boldsymbol{p}_t - proj_{\mathcal{C}}(\boldsymbol{p}_t - \eta_t \nabla \mathcal{L}(\boldsymbol{p}_t))\right)$ and $M = \frac{\mathcal{L}(\boldsymbol{p}_1) - \mathcal{L}^*}{\gamma k} + \frac{\xi^{1/3}\sigma^2}{k} + 2c^2 \sigma^2 k^3 \log(\xi + T)$, where $\sigma^2$ is an abbreviation of the upper bound in Assumption 2.*

**Remark 3.** *In order to obtain an $\epsilon$-solution ($\frac{1}{T} \sum_{t=1}^{T} \mathbb{E}\|G_{\mathcal{C}}(\boldsymbol{p}_t, \gamma)\| \leq \epsilon$), we need to choose $T = \frac{1}{\epsilon^3}$ $I = O(1)$. Thus the total oracle complexity is $O(\frac{1}{\epsilon^3})$.*

**Remark 4.** *The theorems demonstrate that the proposed framework, with variance-reduction techniques and momentum-based updates, ensures convergence towards a prompt distribution that minimizes the empirical risk of the pairwise AUC loss. This implies that the learned prompts are expected to be optimal in terms of performance for the downstream task under the given black-box constraints.*

## 5 Experiments

In this section, we present the experiment setups and provide the results and analysis of both the main experiments and ablation studies. Due to space constraints, additional experimental results are provided in the Appendix D.

## 5.1 Experiment Setups

**Datasets.** To evaluate the effectiveness of our methods, we conduct experiments on 4 datasets including 2 widely used datasets from the GLUE benchmark (Wang et al., 2018), CoLA (Warstadt et al., 2018), MRPC (Dolan & Brockett, 2005),and 2 real-world class imbalanced datasets: Amazon books and electronics reviews (McAuley et al., 2015). For GLUE benchmark, we perform down-sampling on datasets with a given imbalanced ratio (negative to positive samples ratio) $\tau = 20, 50$ to construct the imbalanced scenarios. For 2 real-world datasets, we down-sample datasets with $\tau = 10$, which facilitates experiments and closely approximates the original imbalance ratio. We use AUC to measure the performance of handling imbalanced data.

**Backbone Models.** We select RoBERTa-large (Liu et al., 2019), GPT2-XL (Radford et al., 2019), Llama3 (AI@Meta, 2024) as our backbone models and conduct experiments separately. These models have approximately 355M, 1.5B, and 8B parameters, respectively.

**Baselines.** We compare our proposed methods with the following black-box prompt learning methods under the same experimental settings: **Manual Prompt** performs the zero-shot evaluation on the LLMs with human-written templates, and the results can serve as initial points. **BBT** optimizes the continuous prompts in a random low-dimensional subspace through covariance matrix adaptation evolution strategy (Sun et al., 2022b). **GAP3** introduces a genetic algorithm that considers the prompt as individual and employs auxiliary LLM to generate discrete prompts from the empty (Zhao et al., 2023). **BDPL** utilizes policy gradients to optimize discrete prompt distribution as mentioned in Section 3.1 (Diao et al., 2022).

**Implementation Details.** The proposed methods and all baselines are implemented using PyTorch and experimented on NVIDIA A40 GPUs with 48 GB memory. For all backbone models, we initialize them with checkpoints provided by the HuggingFace. The details of the input template, output label words, and hyperparameters can be found in the Appendix C.

## 5.2 Main Results and Analysis

**Comparison on constructed imbalanced scenarios.** We report the average AUC scores on CoLA over 3 random seeds in Table 1. The results on MRPC can be found in Appendix D.1. and our methods exhibit higher performance compared to all baselines. And in many cases, there are significant improvements. In particular, our methods achieve enhancements over BDPL, confirming that minimizing the pairwise AUC loss can effectively address the class imbalance problem. On the other hand, BBT, GAP3, and BDPL have almost no improvement compared to Manual Prompt, and even there are substantial declines in some cases. We attribute this phenomenon to the fact that these baselines do not have additional handling for imbalanced data. For instance, BDPL uses a cross-entropy loss function, which in imbalanced scenarios leads to the minority class having almost no effect on the training process (Liu et al., 2020).

Table 1: Comparison of AUC scores (mean±std.) on constructed imbalanced scenarios of CoLA. We conduct three groups of experiments on pre-trained RoBERTa-large, GPT2-XL, and Llama3 with a prompt length of 20. The best results are highlighted in **bold**.

| Imbalanced Ratio | Method | RoBERTa-large | GPT2-XL | Llama3 |
|---|---|---|---|---|
| $\tau = 20$ | Manual Prompt | .4586±.0947 | .5224±.0180 | .4917±.0821 |
| | BBT | .4797±.1040 | .5000±.0000 | .4990±.0063 |
| | GAP3 | .5042±.0171 | .5094±.0162 | .5089±.0181 |
| | BDPL | .4880±.0316 | .4963±.0253 | .5193±.0171 |
| | mDP-DPG (ours) | **.5615±.0486** | **.5271±.0064** | **.5453±.0906** |
| $\tau = 50$ | Manual Prompt | .5288±.0481 | .5300±.0017 | .5289±.0501 |
| | BBT | .4094±.0472 | .4938±.0016 | .5111±.0499 |
| | GAP3 | .4944±.0035 | .4989±.0019 | .4983±.0017 |
| | BDPL | .4871±.0105 | .5394±.1131 | .5139±.0580 |
| | mDP-DPG (ours) | **.5700±.0351** | **.5589±.0139** | **.5466±.1314** |

**Comparison on real-world imbalanced datasets.** We conduct experiments with different prompt lengths and report the average AUC scores over 3 different seeds in Table 2. It should be noted that

Table 2: Comparison of AUC values (mean±std.) on real-world imbalanced datasets Amazon books and electronics based on 3 backbone models. $len$ represents the prompt length. OOM represents out-of-memory.

| Model | Method | Book | | Elec | |
|---|---|---|---|---|---|
| | | $len = 20$ | $len = 50$ | $len = 20$ | $len = 50$ |
| RoBERTa -large | Manual Prompt | .8491±.0038 | .8491±.0038 | .8225±.0061 | .8225±.0061 |
| | BBT | .8525±.0032 | .8514±.0065 | .8098±.0172 | .8480±.0348 |
| | GAP3 | .8372±.0115 | .8372±.0115 | .6581±.0239 | .6581±.0239 |
| | BDPL | .8628±.0066 | .8611±.0174 | .8431±.0147 | .8559±.0206 |
| | mDP-DPG (ours) | **.8678±.0084** | **.8623±.0047** | **.8569±.0365** | **.8588±.0289** |
| GPT2-XL | Manual Prompt | .7377±.0068 | .7377±.0068 | .6696±.0544 | .6696±.0544 |
| | BBT | .7406±.0133 | .6078±.0172 | .6284±.0450 | .5196±.0162 |
| | GAP3 | .7785±.0661 | .7785±.0661 | .5459±.0270 | .5459±.0270 |
| | BDPL | .7884±.0475 | .7276±.0040 | .6941±.0256 | .7343±.0295 |
| | mDP-DPG (ours) | **.8721±.0297** | **.7931±.0520** | **.7157±.0384** | **.7353±.0389** |
| Llama3 | Manual Prompt | .7502±.0072 | .7502±.0072 | .6549±.0446 | .6549±.0446 |
| | BBT | .5164±.0142 | .5283±.0127 | .5137±.0221 | .5275±.0119 |
| | GAP3 | OOM | OOM | OOM | OOM |
| | BDPL | .7858±.0363 | .8009±.0179 | .5216±.0162 | .5422±.0427 |
| | mDP-DPG (ours) | **.8098±.0129** | **.8151±.0376** | **.6804±.0814** | **.6657±.1015** |

since GAP3 generates prompts from empty, under our query limit (in Appendix C.2), the lengths of generated prompts are always less than 20. Therefore, the results are the same for maximum prompt lengths of 20 and 50. We have observed that mDP-DPG surpasses all other baselines, which demonstrates our methods remain effective on real-world data. It is worth noting that in some results, mDP-DPG significantly outperforms all baselines, such as the results in the Book with the GPT2-XL. This demonstrates the potential of our method to significantly enhance performance on real-world imbalanced data distributions. Additionally, BBT, GAP3, and BDPL exhibit much poorer performance than Manual Prompt in many results, confirming the deficiencies of these methods in handling imbalanced data.

Table 3: Comparison of AUC values on 3 backbone models. "len" represents the prompt length. $\tau$ denotes the imbalanced ratio (negative to positive samples ratio).

| Model | Method | CoLA (len=20) | | Book ($\tau = 10$) | |
|---|---|---|---|---|---|
| | | $\tau = 20$ | $\tau = 50$ | len=20 | len=50 |
| RoBERTa -large | BDPL-oversample | .4474±.0681 | .4706±.0883 | .8541±.0466 | .8479±.0600 |
| | BDPL-reweight | .5083±.0033 | .4706±.0883 | .8370±.0096 | .8221±.0034 |
| | mDP-DPG(ours) | **.5615±.0486** | **.5700±.0351** | **.8678±.0084** | **.8623±.0047** |
| GPT2-XL | BDPL-oversample | .5172±.0596 | .5156±.0706 | .8171±.0386 | .7413±.0731 |
| | BDPL-reweight | .5057±.0213 | .5428±.0634 | .8290±.0367 | .7628±.0138 |
| | mDP-DPG(ours) | **.5271±.0064** | **.5589±.0139** | **.8721±.0297** | **.7931±.0520** |
| Llama3 | BDPL-oversample | .4943±.0709 | .5333±.0076 | .7826±.0533 | .7699±.0364 |
| | BDPL-reweight | .5151±.0765 | .5082±.0467 | .7973±.0126 | .7169±.0471 |
| | mDP-DPG(ours) | **.5453±.0906** | **.5466±.1314** | **.8098±.0129** | **.8151±.0376** |

**Comparison with Simple Techniques.** In handling imbalanced data, simple techniques like over-sampling minority class samples are among the solutions. To demonstrate the superiority of our proposed methods on imbalanced datasets, we have included such techniques as baselines for comparison. Consequently, we enhance the BDPL approach by incorporating over-sampling and reweighting. We augment the BDPL method as baseline because it formulates black-box prompt learning as a distribution optimization problem and updates the distribution using policy gradients similar to our methods. The results are presented in Table 3. The experimental results demonstrate that our methods outperform simple techniques for handling imbalanced data.

## 5.3 Ablation Study

**Ablation study about the gradient estimator.** To lead to a more stable optimization process, we introduce the variance reduction technique and an unbiased correction term into the gradient estimator. As shown in Figure 2, we provide the performance comparison figure after removing both components on CoLA ($\tau = 20$) and Amazon books with a prompt length of 20. The VR-DPGE gradient estimator exhibits even stronger performance on 3 backbone models. These results indicate that the incorporation of both components in the gradient estimator allows for a more accurate estimation of the gradient.

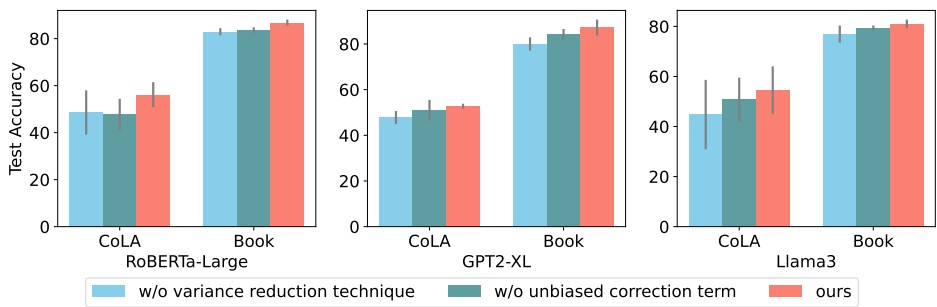

Figure 2: Ablations of variance reduction and unbiased correction term in VR-DPGE.

**Ablation study about the loss function.** We incorporate a hinge loss and compare the results with those obtained using the square loss. The results in Figure 3 indicate that the experimental performance generally decreased with the hinge loss. We believe this is because the square loss is statistically consistent with AUC when used as the surrogate loss, whereas the hinge loss does not have this property (Gao & Zhou, 2012).

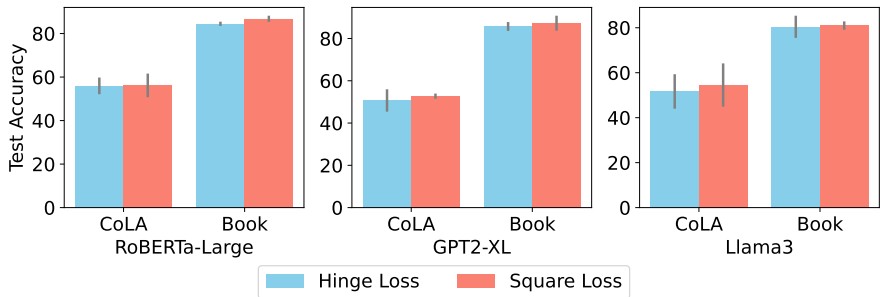

Figure 3: Ablations of loss function. Hinge loss vs Square loss.

## 6 Conclusion

In this paper, we propose a momentum-based imbalanced black-box discrete prompt learning framework mDP-DPG to handle imbalanced data in downstream tasks. Within this framework, we propose VR-DPGE and introduce the STORM technique for variance reduction to achieve more stable optimization. We demonstrate the effectiveness mDP-DPG on constructed imbalanced scenarios and real-world imbalanced datasets, showing performance improvements in class imbalance problems. Although the AUC loss in our framework is specifically tailored for binary classification, we discuss in Appendix D.2 how to overcome the limitations of binary classification and provide additional experimental results. In addition, minimizing pairwise AUC loss in our framework suffers from the challenge of constructing sample pairs from opposite classes. Formulating AUC maximization as an equivalent saddle point problem has become dominant in addressing this challenge. However, this technique cannot be directly applied to our problem, as our objective function requires taking the expectation over prompt $\mathbf{T}$, which would invalidate the existing theoretical derivations. In future work, we will investigate how to introduce it to our framework.

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

# APPENDIX

## A  PROOFS FOR SECTION 3.2

We now prove the explicit derivation for the policy gradient estimator. The $j$-th component of $\nabla_{\boldsymbol{p}_i} \log P(t_i)$ is:

$$\nabla_{\boldsymbol{p}_{i,j}} \log P(t_i) = \nabla_{\boldsymbol{p}_{i,j}} \log \boldsymbol{p}_{i,j_i}.$$

When $j = j_i$, we have

$$\nabla_{\boldsymbol{p}_{i,j}} \log P(t_i) = \nabla_{\boldsymbol{p}_{i,j}} \log \boldsymbol{p}_{i,j_i} = \frac{\nabla_{\boldsymbol{p}_{i,j}} \boldsymbol{p}_{i,j_i}}{\boldsymbol{p}_{i,j_i}} \Big|_{j=j_i} = \frac{1}{\boldsymbol{p}_{i,j_i}}.$$

When $j \neq j_i$, we have

$$\nabla_{\boldsymbol{p}_{i,j}} \log P(t_i) = \nabla_{\boldsymbol{p}_{i,j}} \log \boldsymbol{p}_{i,j_i} = \frac{\nabla_{\boldsymbol{p}_{i,j}} \boldsymbol{p}_{i,j_i}}{\boldsymbol{p}_{i,j_i}} \Big|_{j\neq j_i} = 0.$$

## B  PROOFS FOR SECTION 4.1

### B.1  PROOF OF LEMMA 1

*Proof.* First, it is easy to show that the expectation of $L_{avg}$ over the sampling of the $I$ prompts is equal to the expected loss with expectation taken over the distribution over prompts:

$$\mathbb{E}_S \mathbb{E}_{\{\mathbf{T}^{(\mathbf{k})}\}_{\mathbf{k}=1}^I} L_{avg} \mathbf{1}_{|\mathcal{W}|} \mathbf{1}_n^\top = \mathbb{E}_S \mathbb{E}_{\{\mathbf{T}^{(\mathbf{k})}\}_{\mathbf{k}=1}^I} \frac{1}{I} \Sigma_{\mathbf{k}} L_B(h([\mathbf{T}^{(\mathbf{k})}, \cdot]), S) \mathbf{1}_{|\mathcal{W}|} \mathbf{1}_n^\top$$

$$= \frac{1}{I} \Sigma_{\mathbf{k}} \mathbb{E}_S \mathbb{E}_{\{\mathbf{T}^{(\mathbf{k})}\}_{\mathbf{k}=1}^I} L_B(h([\mathbf{T}^{(\mathbf{k})}, \cdot]), S) \mathbf{1}_{|\mathcal{W}|} \mathbf{1}_n^\top$$

$$= \frac{1}{I} \Sigma_{\mathbf{k}} \mathbb{E}_S \mathbb{E}_{\mathbf{T}} L_B(h([\mathbf{T}, \cdot]), S) \mathbf{1}_{|\mathcal{W}|} \mathbf{1}_n^\top = \frac{I}{I} \mathbb{E}_S \mathbb{E}_{\mathbf{T}} L_B(h([\mathbf{T}, \cdot]), S) \mathbf{1}_{|\mathcal{W}|} \mathbf{1}_n^\top$$

$$= \mathbb{E}_S \mathbb{E}_{\mathbf{T}} L_B(h([\mathbf{T}, \cdot]), S) \mathbf{1}_{|\mathcal{W}|} \mathbf{1}_n^\top \qquad (11)$$

Now, let us analyze the full expression for the expectation of $\boldsymbol{g_p}$:

$$\mathbb{E}_S \mathbb{E}_{\{\mathbf{T}^{(\mathbf{k})}\}_{\mathbf{k}=1}^I} \boldsymbol{g_p} = \mathbb{E}_S \mathbb{E}_{\{\mathbf{T}^{(\mathbf{k})}\}_{\mathbf{k}=1}^I} \left( L_{avg} \mathbf{1}_{|\mathcal{W}|} \mathbf{1}_n^\top + \frac{1}{I-1} \Sigma_k \nabla \log P(\mathbf{T}^{(\mathbf{k})})(L_B(h([\mathbf{T}^{(\mathbf{k})}, \cdot]), S) - L_{avg}) \right)$$

$$= \mathbb{E}_S \mathbb{E}_{\{\mathbf{T}^{(\mathbf{k})}\}_{\mathbf{k}=1}^I} L_{avg} \mathbf{1}_{|\mathcal{W}|} \mathbf{1}_n^\top + \mathbb{E}_S \mathbb{E}_{\{\mathbf{T}^{(\mathbf{k})}\}_{\mathbf{k}=1}^I} \underbrace{\left( \frac{1}{I-1} \Sigma_k \nabla \log P(\mathbf{T}^{(\mathbf{k})})(L_B(h([\mathbf{T}^{(\mathbf{k})}, \cdot]), S) - L_{avg}) \right)}_{\text{(A)}}$$

$$(12)$$

We can rewrite Ⓐ as follows:

$$\text{Ⓐ} = \frac{1}{I-1} \sum_{\mathbf{k}=1}^I \left( L_B(h([\mathbf{T}^{(\mathbf{k})}, \cdot]), S) - \frac{1}{I} \sum_{\mathbf{j}=1}^I L_B(h([\mathbf{T}^{(\mathbf{j})}, \cdot]), S) \right) \nabla_{\boldsymbol{p}} \log P(\mathbf{T}^{(\mathbf{k})})$$

$$= \frac{1}{I-1} \sum_{\mathbf{k}=1}^I \left( \frac{1}{I} \sum_{\mathbf{j}=1}^I \left( L_B(h([\mathbf{T}^{(\mathbf{k})}, \cdot]), S) - L_B(h([\mathbf{T}^{(\mathbf{j})}, \cdot]), S) \right) \right) \nabla_{\boldsymbol{p}} \log P(\mathbf{T}^{(\mathbf{k})})$$

$$= \frac{1}{I-1} \sum_{\mathbf{k}=1}^I \left( \frac{1}{I} \sum_{\mathbf{j}=1, \mathbf{j} \neq \mathbf{k}}^I \left( L_B(h([\mathbf{T}^{(\mathbf{k})}, \cdot]), S) - L_B(h([\mathbf{T}^{(\mathbf{j})}, \cdot]), S) \right) \right) \nabla_{\boldsymbol{p}} \log P(\mathbf{T}^{(\mathbf{k})})$$

$$= \underbrace{\frac{1}{I} \sum_{\mathbf{k}=1}^I L_B(h([\mathbf{T}^{(\mathbf{k})}, \cdot]), S) \nabla_{\boldsymbol{p}} \log P(\mathbf{T}^{(\mathbf{k})})}_{\text{①}} - \underbrace{\frac{1}{I} \sum_{\mathbf{k}=1}^I \frac{1}{I-1} \sum_{\mathbf{j} \neq \mathbf{k}}^I L_B(h([\mathbf{T}^{(\mathbf{j})}, \cdot]), S) \nabla_{\boldsymbol{p}} \log P(\mathbf{T}^{(\mathbf{k})}))}_{\text{②}}$$

Now, first, we have:

$$\mathbb{E}_S\mathbb{E}_{\{\mathbf{T^{(k)}}\}_{\mathbf{k}=1}^I}\left[\boxed{1}\right] = \mathbb{E}_S\mathbb{E}_{\{\mathbf{T^{(k)}}\}_{\mathbf{k}=1}^I}\frac{1}{I}\sum_{\mathbf{k}=1}^I L_B(h([\mathbf{T^{(k)}},\cdot]),S)\nabla_{\boldsymbol{p}}\log P(\mathbf{T^{(k)}})$$

$$= \mathbb{E}_S\frac{1}{I}\sum_{\mathbf{k}=1}^I \mathbb{E}_{\{\mathbf{T^{(k)}}\}_{\mathbf{k}=1}^I} L_B(h([\mathbf{T^{(k)}},\cdot]),S)\frac{\nabla_{\boldsymbol{p}}P(\mathbf{T^{(k)}})}{P(\mathbf{T^{(k)}})}$$

$$= \mathbb{E}_S\frac{1}{I}\sum_{\mathbf{k}=1}^I \mathbb{E}_{\mathbf{T^{(k)}}} L_B(h([\mathbf{T^{(k)}},\cdot]),S)\frac{\nabla_{\boldsymbol{p}}P(\mathbf{T^{(k)}})}{P(\mathbf{T^{(k)}})}$$

$$= \mathbb{E}_S\frac{1}{I}\sum_{\mathbf{k}=1}^I \sum_{\mathbf{T^{(k)}}} P(\mathbf{T^{(k)}})L_B(h([\mathbf{T^{(k)}},\cdot]),S)\frac{\nabla_{\boldsymbol{p}}P(\mathbf{T^{(k)}})}{P(\mathbf{T^{(k)}})}$$

$$= \mathbb{E}_S\frac{1}{I}\sum_{\mathbf{k}=1}^I \sum_{\mathbf{T^{(k)}}} L_B(h([\mathbf{T^{(k)}},\cdot]),S)\nabla_{\boldsymbol{p}}P(\mathbf{T^{(k)}})$$

$$= \nabla_{\boldsymbol{p}}\mathbb{E}_S\frac{1}{I}\sum_{\mathbf{k}=1}^I \sum_{\mathbf{T^{(k)}}} P(\mathbf{T^{(k)}})L_B(h([\mathbf{T^{(k)}},\cdot]),S)$$

$$= \nabla_{\boldsymbol{p}}\mathbb{E}_S\frac{1}{I}\sum_{\mathbf{k}=1}^I \mathbb{E}_{\mathbf{T^{(k)}}} L_B(h([\mathbf{T^{(k)}},\cdot]),S)$$

$$= \nabla_{\boldsymbol{p}}\mathbb{E}_S\frac{I}{I}\mathbb{E}_{\mathbf{T}} L_B(h([\mathbf{T},\cdot]),S) = \nabla_{\boldsymbol{p}}\mathbb{E}_S\mathbb{E}_{\mathbf{T}} L_B(h([\mathbf{T},\cdot]),S) \quad (13)$$

Then, we have:

$$\mathbb{E}_S\mathbb{E}_{\{\mathbf{T^{(k)}}\}_{\mathbf{k}=1}^I}\left[\boxed{2}\right] = \mathbb{E}_S\mathbb{E}_{\{\mathbf{T^{(k)}}\}_{\mathbf{k}=1}^I}\frac{1}{I}\sum_{\mathbf{k}=1}^I \frac{1}{I-1}\sum_{\mathbf{j}\neq\mathbf{k}}^I L_B(h([\mathbf{T^{(j)}},\cdot]),S)\nabla_{\boldsymbol{p}}\log P(\mathbf{T^{(k)}})$$

$$= \mathbb{E}_S\frac{1}{I}\sum_{\mathbf{k}=1}^I \mathbb{E}_{\{\mathbf{T^{(k)}}\}_{\mathbf{k}=1}^I}\frac{1}{I-1}\sum_{\mathbf{j}\neq\mathbf{k}}^I L_B(h([\mathbf{T^{(j)}},\cdot]),S)\nabla_{\boldsymbol{p}}\log P(\mathbf{T^{(k)}})$$

$$= \mathbb{E}_S\frac{1}{I}\sum_{\mathbf{k}=1}^I \mathbb{E}_{\mathbf{T^{(k)}}}\mathbb{E}_{\{\mathbf{T^{(j)}}\}_{\mathbf{j}=1,\mathbf{j}\neq\mathbf{k}}^I}\frac{1}{I-1}\sum_{\mathbf{j}\neq\mathbf{k}}^I L_B(h([\mathbf{T^{(j)}},\cdot]),S)\nabla_{\boldsymbol{p}}\log P(\mathbf{T^{(k)}})$$

$$= \mathbb{E}_S\frac{1}{I}\sum_{\mathbf{k}=1}^I \frac{1}{I-1}\mathbb{E}_{\mathbf{T^{(k)}}}\sum_{\mathbf{j}\neq\mathbf{k}}^I \mathbb{E}_{\{\mathbf{T^{(j)}}\}_{\mathbf{j}=1,\mathbf{j}\neq\mathbf{k}}^I} L_B(h([\mathbf{T^{(j)}},\cdot]),S)\nabla_{\boldsymbol{p}}\log P(\mathbf{T^{(k)}})$$

$$= \mathbb{E}_S\frac{1}{I}\sum_{\mathbf{k}=1}^I \frac{1}{I-1}\mathbb{E}_{\mathbf{T^{(k)}}}\sum_{\mathbf{j}\neq\mathbf{k}}^I \mathbb{E}_{\mathbf{T^{(j)}}} L_B(h([\mathbf{T^{(j)}},\cdot]),S)\nabla_{\boldsymbol{p}}\log P(\mathbf{T^{(k)}})$$

$$= \mathbb{E}_S\frac{1}{I}\sum_{\mathbf{k}=1}^I \frac{1}{I-1}\sum_{\mathbf{j}\neq\mathbf{k}}^I \mathbb{E}_{\mathbf{T^{(j)}}} L_B(h([\mathbf{T^{(j)}},\cdot]),S)\mathbb{E}_{\mathbf{T^{(k)}}}\nabla_{\boldsymbol{p}}\log P(\mathbf{T^{(k)}})$$

$$= \mathbb{E}_S\frac{1}{I}\sum_{\mathbf{k}=1}^I \frac{1}{I-1}\sum_{\mathbf{j}\neq\mathbf{k}}^I \mathbb{E}_{\mathbf{T^{(j)}}} L_B(h([\mathbf{T^{(j)}},\cdot]),S)\mathbb{E}_{\mathbf{T^{(k)}}}\frac{\nabla_{\boldsymbol{p}}P(\mathbf{T^{(k)}})}{P(\mathbf{T^{(k)}})}$$

$$= \mathbb{E}_S\frac{1}{I}\sum_{\mathbf{k}=1}^I \frac{1}{I-1}\sum_{\mathbf{j}\neq\mathbf{k}}^I \mathbb{E}_{\mathbf{T^{(j)}}} L_B(h([\mathbf{T^{(j)}},\cdot]),S)\sum_{\mathbf{T^{(k)}}} P(\mathbf{T^{(k)}})\frac{\nabla_{\boldsymbol{p}}P(\mathbf{T^{(k)}})}{P(\mathbf{T^{(k)}})}$$

$$= \mathbb{E}_S\frac{1}{I}\sum_{\mathbf{k}=1}^I \frac{1}{I-1}\sum_{\mathbf{j}\neq\mathbf{k}}^I \mathbb{E}_{\mathbf{T^{(j)}}} L_B(h([\mathbf{T^{(j)}},\cdot]),S)\sum_{\mathbf{T^{(k)}}} \nabla_{\boldsymbol{p}}P(\mathbf{T^{(k)}})$$

$$= \mathbb{E}_S \frac{1}{I} \sum_{\mathbf{k}=1}^{I} \frac{1}{I-1} \sum_{\mathbf{j}\neq\mathbf{k}}^{I} \mathbb{E}_{\mathbf{T}^{(\mathbf{j})}} L_B(h([\mathbf{T}^{(\mathbf{j})},\cdot]),S) \nabla_{\boldsymbol{p}} \sum_{\mathbf{T}^{(\mathbf{k})}} P(\mathbf{T}^{(\mathbf{k})}) \tag{14}$$

Now, for any $i \in [n]$, we have:

$$\nabla_{\boldsymbol{p}_i} \sum_{\mathbf{T}^{(\mathbf{k})}} P(\mathbf{T}^{(\mathbf{k})})) = \nabla_{\boldsymbol{p}_i} \sum_{t_1^{(\mathbf{k})}\in\mathcal{W}} ... \sum_{t_i^{(\mathbf{k})}\in\mathcal{W}} ... \sum_{t_n^{(\mathbf{k})}\in\mathcal{W}} P(t_1^{(\mathbf{k})})...P(t_i^{(\mathbf{k})})...P(t_n^{(\mathbf{k})})$$

$$= \nabla_{\boldsymbol{p}_i} \sum_{t_i^{(\mathbf{k})}\in\mathcal{W}} P(t_i^{(\mathbf{k})}) \left( \sum_{t_1^{(\mathbf{k})}\in\mathcal{W}} P(t_1^{(\mathbf{k})}) \left( \sum_{t_2^{(\mathbf{k})}\in\mathcal{W}} P(t_2^{(\mathbf{k})}) \left( ... \left( \sum_{t_{i-1}^{(\mathbf{k})}\in\mathcal{W}} P(t_{i+1}^{(\mathbf{k})}) \left( ... \left( \sum_{t_n^{(\mathbf{k})}\in\mathcal{W}} P(t_n^{(\mathbf{k})}) \right) \right) \right) \right) \right) \right)$$

$$= \nabla_{\boldsymbol{p}_i} \sum_{t_i^{(\mathbf{k})}\in\mathcal{W}} P(t_i^{(\mathbf{k})}) = \sum_{t_i^{(\mathbf{k})}\in\mathcal{W}} \nabla_{\boldsymbol{p}_i} P(t_i^{(\mathbf{k})}) = \sum_{t_i^{(\mathbf{k})}\in\mathcal{W}} \begin{bmatrix} 0 \\ \vdots \\ 1 \\ \vdots \\ 0 \end{bmatrix} \leftarrow \text{index } j \text{ s.t. the } j\text{-th word from } \mathcal{W} \text{ is } t_i^{(\mathbf{k})} = \begin{bmatrix} 1 \\ \vdots \\ 1 \\ \vdots \\ 1 \end{bmatrix}$$

Now, plugging the result above into 14, we obtain:

$$\mathbb{E}_S \mathbb{E}_{\{\mathbf{T}^{(\mathbf{k})}\}_{\mathbf{k}=1}^{I}} \left[ \textcircled{2} \right] = \mathbb{E}_S \frac{1}{I} \sum_{\mathbf{k}=1}^{I} \frac{1}{I-1} \sum_{\mathbf{j}\neq\mathbf{k}}^{I} \mathbb{E}_{\mathbf{T}^{(\mathbf{j})}} L_B(h([\mathbf{T}^{(\mathbf{j})},\cdot]),S) \mathbf{1}_{|\mathcal{W}|} \mathbf{1}_n^\top,$$

Continuing from 14, since the term in the sum in 14 is a constant (as for all $j$, the $\mathbf{T}_j^{(\mathbf{k})}$ are sampled i.i.d):

$$\mathbb{E}_S \mathbb{E}_{\{\mathbf{T}^{(\mathbf{k})}\}_{\mathbf{k}=1}^{I}} \left[ \textcircled{2} \right] = \mathbb{E}_S \frac{I}{I} \frac{I-1}{I-1} \mathbb{E}_{\mathbf{T}} L_B(h([\mathbf{T},\cdot]),S) \mathbf{1}_{|\mathcal{W}|} \mathbf{1}_n^\top$$
$$= \mathbb{E}_S \mathbb{E}_{\mathbf{T}} L_B(h([\mathbf{T},\cdot]),S) \mathbf{1}_{|\mathcal{W}|} \mathbf{1}_n^\top \tag{15}$$

Therefore, plugging 11, 13 and 15 into 12, we obtain:

$$\mathbb{E}_S \mathbb{E}_{\{\mathbf{T}^{(\mathbf{k})}\}_{\mathbf{k}=1}^{I}} \boldsymbol{g}_{\boldsymbol{p}} = \mathbb{E}_S \mathbb{E}_{\mathbf{T}} L_B(h([\mathbf{T},\cdot]),S) \mathbf{1}_{|\mathcal{W}|} \mathbf{1}_n^\top + \nabla_{\boldsymbol{p}} \mathbb{E}_S \mathbb{E}_{\mathbf{T}} L_B(h([\mathbf{T},\cdot]),S)$$
$$- \mathbb{E}_S \mathbb{E}_{\mathbf{T}} L_B(h([\mathbf{T},\cdot]),S) \mathbf{1}_{|\mathcal{W}|} \mathbf{1}_n^\top$$
$$= \nabla_{\boldsymbol{p}} \mathbb{E}_S \mathbb{E}_{\mathbf{T}} L_B(h([\mathbf{T},\cdot]),S)$$

$\square$

### B.2 PROOF OF LEMMA 2

*Proof.* Let $\boldsymbol{p} \in \mathcal{C}$. Let us denote by $\mathbb{E}_{\mathbf{T}} := \mathbb{E}_{\mathbf{T}\sim\mathcal{P}_{\boldsymbol{p}}}$ for simplicity. We can express the gradient estimate as follows:

$$\nabla_{\boldsymbol{p}} \mathbb{E}_S \mathbb{E}_{\mathbf{T}} L_B(h([\mathbf{T},\cdot]),S) = \nabla_{\boldsymbol{p}} \mathbb{E}_S \mathbb{E}_{j_1\sim\text{Cat}(\boldsymbol{p}_1),...,j_n\sim\text{Cat}(\boldsymbol{p}_n)} L_B(h([t_{j_1},...,t_{j_n},\cdot]),S)$$
$$= \mathbb{E}_S \nabla_{\boldsymbol{p}} \sum_{j_1\in[|\mathcal{W}|]} ... \sum_{j_n\in[|\mathcal{W}|]} L_B(h([t_{j_1},...,t_{j_n},\cdot]),S) \Pi_{i=1}^n \boldsymbol{p}_{j_i,i}$$
$$= \mathbb{E}_S \sum_{j_1\in[|\mathcal{W}|]} ... \sum_{j_n\in[|\mathcal{W}|]} L_B(h([t_{j_1},...,t_{j_n},\cdot]),S) \nabla_{\boldsymbol{p}} \Pi_{i=1}^n \boldsymbol{p}_{j_i,i}$$

Therefore:

$$\nabla_{\boldsymbol{p}_{k,l}} \mathbb{E}_S \mathbb{E}_{\mathbf{T}} L_B(h([\mathbf{T},\cdot]),S) = \mathbb{E}_S \sum_{j_1\in[|\mathcal{W}|]} ... \sum_{j_n\in[|\mathcal{W}|]} L_B(h([t_{j_1},...,t_{j_n},\cdot]),S) \nabla_{\boldsymbol{p}_{k,l}} \Pi_{i=1}^n \boldsymbol{p}_{j_i,i}$$

Now, we have:

$$\nabla_{\boldsymbol{p}_{k,l}}\Pi_{i=1}^{n}\boldsymbol{p}_{j_i,i} = \begin{cases} \Pi_{i=1,i\neq l}^{n}\boldsymbol{p}_{j_i,i} \text{ if } j_l = k \\ 0 \text{ otherwise} \end{cases}$$

Therefore:

$$\nabla_{\boldsymbol{p}_{k,l}}\mathbb{E}_S\mathbb{E}_{\mathbf{T}}L_B(h([\mathbf{T},\cdot]),S)$$

$$= \mathbb{E}_S \sum_{j_1\in[|\mathcal{W}|]} ... \sum_{j_n\in[|\mathcal{W}|]} L_B(h([t_{j_1},...,t_{j_n},\cdot]),S)\nabla_{\boldsymbol{p}_{k,l}}\Pi_{i=1}^{n}\boldsymbol{p}_{j_i,i}$$

$$= \mathbb{E}_S \sum_{j_1\in[|\mathcal{W}|]} ... \sum_{j_{l-1}\in[|\mathcal{W}|]} \sum_{j_{l+1}\in[|\mathcal{W}|]} ... \sum_{j_n\in[|\mathcal{W}|]} L_B(h([t_{j_1},...,t_{l-1},t_k,t_{l+1},...,t_{j_n},\cdot]),S)\Pi_{i=1,i\neq l}^{n}\boldsymbol{p}_{j_i,i}$$

Note that the last expression above can be expressed as an expectation, in the case where each column of $\boldsymbol{p}$ defines a probability distribution:

$$\nabla_{\boldsymbol{p}_{k,l}}\mathbb{E}_S\mathbb{E}_{\mathbf{T}}L_B(h([\mathbf{T},\cdot]),S) = \mathbb{E}_S\mathbb{E}_{j_1,...,j_{l-1},j_{l+1}...,j_n}L_B(h([t_{j_1},...,t_{l-1},t_k,t_{l+1},...,t_{j_n},\cdot]),S)$$

Similarly, we can compute the Hessian of such cost function:

$$\frac{\partial^2}{\partial\boldsymbol{p}_{k,l}\partial\boldsymbol{p}_{m,q}}\mathbb{E}_S\mathbb{E}_{\mathbf{T}}L_B(h([\mathbf{T},\cdot]),S)$$

$$= \mathbb{E}_S\frac{\partial}{\partial\boldsymbol{p}_{m,q}}\left[ \sum_{j_1\in[|\mathcal{W}|]} ... \sum_{j_{l-1}\in[|\mathcal{W}|]} \sum_{j_{l+1}\in[|\mathcal{W}|]} ... \sum_{j_n\in[|\mathcal{W}|]} L_B(h([t_{j_1},...,t_{l-1},t_k,t_{l+1},...,t_{j_n},\cdot]),S)\Pi_{i=1,i\neq l}^{n}\boldsymbol{p}_{j_i,i} \right]$$

$$= \mathbb{E}_S\left[ \sum_{j_1\in[|\mathcal{W}|]} ... \sum_{j_{l-1}\in[|\mathcal{W}|]} \sum_{j_{l+1}\in[|\mathcal{W}|]} ... \sum_{j_n\in[|\mathcal{W}|]} L_B(h([t_{j_1},...,t_{j_n},\cdot]),S)\frac{\partial}{\partial\boldsymbol{p}_{m,q}}\Pi_{i=1,i\neq l}^{n}\boldsymbol{p}_{j_i,i} \right]$$

Now, similarly as before, we have:

$$\frac{\partial}{\partial\boldsymbol{p}_{m,q}}\Pi_{i=1,i\neq l}^{n}\boldsymbol{p}_{j_i,i} = \begin{cases} \Pi_{i=1,i\neq l,i\neq k}^{n}\boldsymbol{p}_{j_i,i} \text{ if } j_q = m \text{ and } q\neq l \\ 0 \text{ otherwise} \end{cases}$$

Therefore, the last expression above can be expressed as an expectation, if each column of $\boldsymbol{p}$ defines a probability distribution:

$$\frac{\partial^2}{\partial\boldsymbol{p}_{k,l}\partial\boldsymbol{p}_{m,n}}\mathbb{E}_S\mathbb{E}_{\mathbf{T}}L_B(h([\mathbf{T},\cdot]),S)$$

$$= \begin{cases} \mathbb{E}_S\mathbb{E}_{j_1,...,j_{l-1},j_{l+1}..j_{q-1},j_{q+1}...,j_n}L_B(h([t_{j_1},...,t_{j_{l-1}},t_k,t_{j_{l+1}},...,t_{j_{q-1}},t_m,t_{j_{q+1}},...,t_{j_n},\cdot],S)) \text{ if } q\neq l \\ 0 \text{ otherwise.} \end{cases}$$

Therefore, using Assumption 1, we have that:

$$\frac{\partial^2}{\partial\boldsymbol{p}_{k,l}\partial\boldsymbol{p}_{m,n}}\mathbb{E}_S\mathbb{E}_{\mathbf{T}}L_B(h([\mathbf{T},\cdot]),S) \leq C$$

which implies, with $H(\boldsymbol{p})$ denoting the Hessian of $\mathbb{E}_S\mathbb{E}_{\mathbf{T}}L_B(h([\mathbf{T},\cdot]),S)$ with respect to $\boldsymbol{p}$:

$$||H(\boldsymbol{p})||_F \leq \sqrt{n|\mathcal{W}|C^2} = \sqrt{n|\mathcal{W}|}C$$

And therefore we have :

$$||H(\boldsymbol{p})||_2 \leq ||H(\boldsymbol{p})||_F \leq \sqrt{n|\mathcal{W}|}C,$$

using (2.3.7) in Golub & Van Loan (2013). We can now use Lemma 1.2.2 in Nesterov et al. (2018) to relate such bound on the Hessian to the smoothness constant of $\mathbb{E}_S \mathbb{E}_\mathbf{T} L_B(h([\mathbf{T}, \cdot]), S)$.

$\square$

## B.3 PROOF OF BOUNDED VARIANCE (REMARK 2)

*Proof.* Consider the following constraints set: $\boldsymbol{\mathcal{C}} = \{\boldsymbol{p} \in \mathbb{R}^{|\mathcal{W}| \times n} : \forall i \in [n], \|\boldsymbol{p}_i\|_1 = 1, \forall j \in [|\mathcal{W}|], \nu \leq \boldsymbol{p}_{j,i} \leq 1\}$, for some $\nu \in (0,1)$. For simplicity, we denote $L(\mathbf{T}^{(\mathbf{k})}) := L_B(h([\mathbf{T}^{(\mathbf{k})}, \cdot]), S)$, and denote $\boldsymbol{e}_i = \begin{bmatrix} 0 \\ \vdots \\ 1 \\ \vdots \\ 0 \end{bmatrix} \leftarrow i$. For any $i \in [n]$, we have:

$$\mathrm{Var}\left(\boldsymbol{g}_{\boldsymbol{p}_i}\right) = \mathrm{Var}\left(L_{avg}\mathbf{1}_{|\mathcal{W}|} + \frac{1}{I-1}\Sigma_k \nabla_{\boldsymbol{p}_i}\log P(t_i^{(\mathbf{k})})(L(\mathbf{T}^{(\mathbf{k})}) - L_{avg})\right)$$

$$= \mathrm{Var}\left(L_{avg}\mathbf{1}_{|\mathcal{W}|} + \frac{1}{I-1}\Sigma_k \frac{\boldsymbol{e}_{j_i^{(\mathbf{k})}}}{P(t_i^{(\mathbf{k})})}(L(\mathbf{T}^{(\mathbf{k})}) - L_{avg})\right)$$

$$\overset{(a)}{\leq} \mathbb{E}\left\|L_{avg}\mathbf{1}_{|\mathcal{W}|} + \frac{1}{I-1}\Sigma_k \frac{\boldsymbol{e}_{j_i^{(\mathbf{k})}}}{P(t_i^{(\mathbf{k})})}(L(\mathbf{T}^{(\mathbf{k})}) - L_{avg})\right\|^2$$

$$= \mathbb{E}\|L_{avg}\mathbf{1}_{|\mathcal{W}|}\|^2 + \mathbb{E}\left\|\frac{1}{I-1}\Sigma_k \frac{\boldsymbol{e}_{j_i^{(\mathbf{k})}}}{P(t_i^{(\mathbf{k})})}(L(\mathbf{T}^{(\mathbf{k})}) - L_{avg})\right\|^2$$

$$\quad + 2\mathbb{E}\langle L_{avg}\mathbf{1}_{|\mathcal{W}|}, \frac{1}{I-1}\Sigma_k \frac{\boldsymbol{e}_{j_i^{(\mathbf{k})}}}{P(t_i^{(\mathbf{k})})}(L(\mathbf{T}^{(\mathbf{k})}) - L_{avg})\rangle$$

$$\overset{(b)}{=} \frac{I}{I^2}\mathbb{E}_\mathbf{T}\|L(\mathbf{T})\mathbf{1}_{|\mathcal{W}|}\|^2 + \frac{I(I-1)}{I^2}\|\mathbb{E}_\mathbf{T}L(\mathbf{T})\mathbf{1}_{|\mathcal{W}|}\|^2 + \mathbb{E}\left\|\frac{1}{I-1}\Sigma_k \frac{\boldsymbol{e}_{j_i^{(\mathbf{k})}}}{P(t_i^{(\mathbf{k})})}(L(\mathbf{T}^{(\mathbf{k})}) - L_{avg})\right\|^2$$

$$\quad + 2\frac{1}{I-1}\Sigma_k\left[\mathbb{E}L_{avg}\left(\frac{L(\mathbf{T}^{(\mathbf{k})}) - L_{avg}}{P(t_i^{(\mathbf{k})})}\right)\langle \mathbf{1}_{|\mathcal{W}|}, \boldsymbol{e}_{j_i^{(\mathbf{k})}}\rangle\right]$$

$$= \frac{|\mathcal{W}|}{I}\mathbb{E}_\mathbf{T}\|L(\mathbf{T})\|^2 + \frac{|\mathcal{W}|(I-1)}{I}\|\mathbb{E}_\mathbf{T}L(\mathbf{T})\|^2 + \mathbb{E}\left\|\frac{1}{I-1}\Sigma_k \frac{\boldsymbol{e}_{j_i^{(\mathbf{k})}}}{P(t_i^{(\mathbf{k})})}(L(\mathbf{T}^{(\mathbf{k})}) - L_{avg})\right\|^2$$

$$\quad + 2\frac{1}{I-1}\Sigma_k\left[\mathbb{E}L_{avg}\left(\frac{L(\mathbf{T}^{(\mathbf{k})}) - L_{avg}}{P(t_i^{(\mathbf{k})})}\right)\right]$$

$$= \frac{|\mathcal{W}|}{I}\mathbb{E}_\mathbf{T}\|L(\mathbf{T})\|^2 + \frac{|\mathcal{W}|(I-1)}{I}\|\mathbb{E}_\mathbf{T}L(\mathbf{T})\|^2 + \mathbb{E}\left\|\frac{1}{I-1}\Sigma_k \frac{\boldsymbol{e}_{j_i^{(\mathbf{k})}}}{P(t_i^{(\mathbf{k})})}(L(\mathbf{T}^{(\mathbf{k})}) - L_{avg})\right\|^2$$

$$\quad + 2\frac{1}{I-1}\Sigma_k\left[\mathbb{E}_{t_1^{(\mathbf{k})},\ldots,t_{i-1}^{(\mathbf{k})},t_{i+1}^{(\mathbf{k})},\ldots,t_n^{(\mathbf{k})}}\sum_{t_i^{(\mathbf{k})}\in\mathcal{W}}P(t_i^{(\mathbf{k})})\frac{1}{P(t_i^{(\mathbf{k})})}L_{avg}\left(L(\mathbf{T}^{(\mathbf{k})}) - L_{avg}\right)\right]$$

$$= \frac{|\mathcal{W}|}{I}\mathbb{E}_{\mathbf{T}}\|L(\mathbf{T})\|^2 + \frac{|\mathcal{W}|(I-1)}{I}\|\mathbb{E}_{\mathbf{T}}L(\mathbf{T})\|^2 + \mathbb{E}\left\|\frac{1}{I-1}\Sigma_k \frac{\boldsymbol{e}_{j_i^{(\mathbf{k})}}}{P(t_i^{(\mathbf{k})})}(L(\mathbf{T}^{(\mathbf{k})}) - L_{avg})\right\|^2$$

$$+ 2\frac{1}{I-1}\Sigma_k \left[\mathbb{E}_{t_1^{(\mathbf{k})},\dots,t_{i-1}^{(\mathbf{k})},t_{i+1}^{(\mathbf{k})},\dots,t_n^{(\mathbf{k})}} \sum_{t_i^{(\mathbf{k})}\in\mathcal{W}} L_{avg}\left(L(\mathbf{T}^{(\mathbf{k})}) - L_{avg}\right)\right]$$

$$\overset{(c)}{\leq} \frac{|\mathcal{W}|}{I}C^2 + \frac{|\mathcal{W}|(I-1)}{I}C^2 + \mathbb{E}\left\|\frac{1}{I-1}\Sigma_k \frac{\boldsymbol{e}_{j_i^{(\mathbf{k})}}}{P(t_i^{(\mathbf{k})})}(L(\mathbf{T}^{(\mathbf{k})}) - L_{avg})\right\|^2 + 2\frac{I}{I-1}|\mathcal{W}|2C^2$$

$$= \frac{|\mathcal{W}|}{I}C^2 + \frac{|\mathcal{W}|(I-1)}{I}C^2$$

$$+ \frac{1}{(I-1)^2}\left(\sum_{\mathbf{k}} \mathbb{E}\left\|\frac{\boldsymbol{e}_{j_i^{(\mathbf{k})}}}{P(t_i^{(\mathbf{k})})}(L(\mathbf{T}^{(\mathbf{k})}) - L_{avg})\right\|^2\right.$$

$$\left. + \sum_{\mathbf{l}}\sum_{\mathbf{m},\mathbf{m}\neq\mathbf{l}} \mathbb{E}\langle \frac{\boldsymbol{e}_{j_i^{(l)}}}{P(t_i^{(\mathbf{l})})}(L(\mathbf{T}^{(\mathbf{l})}) - L_{avg}), \frac{\boldsymbol{e}_{j_i^{(m)}}}{P(t_i^{(\mathbf{m})})}(L(\mathbf{T}^{(\mathbf{m})}) - L_{avg})\rangle\right) + 2\frac{I}{I-1}|\mathcal{W}|2C^2$$

$$\overset{(d)}{\leq} \frac{|\mathcal{W}|}{I}C^2 + \frac{|\mathcal{W}|(I-1)}{I}C^2 + \frac{1}{(I-1)^2}\left(\sum_{\mathbf{k}} \mathbb{E}\left(\frac{1}{P(t_i^{(\mathbf{k})})}(L(\mathbf{T}^{(\mathbf{k})}) - L_{avg})\right)^2\right.$$

$$+ \mathbb{E}\sum_{\mathbf{l}}\sum_{\mathbf{m},\mathbf{m}\neq\mathbf{l}} \left\|\frac{\boldsymbol{e}_{j_i^{(l)}}}{P(t_i^{(\mathbf{l})})}(L(\mathbf{T}^{(\mathbf{l})}) - L_{avg})\right\|\left\|\frac{\boldsymbol{e}_{j_i^{(m)}}}{P(t_i^{(\mathbf{m})})}(L(\mathbf{T}^{(\mathbf{m})}) - L_{avg})\right\|\right) + 2\frac{I}{I-1}|\mathcal{W}|2C^2$$

$$= \frac{|\mathcal{W}|}{I}C^2 + \frac{|\mathcal{W}|(I-1)}{I}C^2 + \frac{1}{(I-1)^2}\left(\sum_{\mathbf{k}} \mathbb{E}_{\{t_p^{(\mathbf{k})}\}_{p=1}^n \setminus t_i^{(\mathbf{k})}} \sum_{t_i^{(\mathbf{k})}\in\mathcal{W}} P(t_i^{(\mathbf{k})})\left(\frac{1}{P(t_i^{(\mathbf{k})})}(L(\mathbf{T}^{(\mathbf{k})}) - L_{avg})\right)^2\right.$$

$$+ \sum_{\mathbf{l}}\sum_{\mathbf{m},\mathbf{m}\neq\mathbf{l}} \mathbb{E}_{\{t_p^{(\mathbf{l})}\}_{p=1}^n \setminus t_i^{(\mathbf{l})}}\mathbb{E}_{\{t_p^{(\mathbf{m})}\}_{p=1}^n \setminus t_i^{(\mathbf{m})}} \sum_{t_i^{(l)}\in\mathcal{W}} P(t_i^{(\mathbf{l})}) \sum_{t_i^{(m)}\in\mathcal{W}} P(t_i^{(\mathbf{m})})\frac{(L(\mathbf{T}^{(\mathbf{l})}) - L_{avg})(L(\mathbf{T}^{(\mathbf{m})}) - L_{avg})}{P(t_i^{(\mathbf{l})})P(t_i^{(\mathbf{m})})}\right)$$

$$+ 2\frac{I}{I-1}|\mathcal{W}|2C^2$$

$$\leq \frac{|\mathcal{W}|}{I}C^2 + \frac{|\mathcal{W}|(I-1)}{I}C^2 + \frac{I|\mathcal{W}|}{\nu(I-1)^2}4C^2 + \frac{1}{(I-1)^2}\left(I(I-1)|\mathcal{W}|^2 4C^2\right) + 2\frac{I}{I-1}|\mathcal{W}|2C^2$$

$$= |\mathcal{W}|C^2 + \frac{I|\mathcal{W}|4C^2}{\nu(I-1)^2} + \frac{I4|\mathcal{W}|^2 C^2}{I-1} + \frac{4C^2|\mathcal{W}|I}{I-1}$$

Where (a) and (b) follow from the fact that for some random variable $X$: $\text{Var}(X) = \mathbb{E}\|X - \mathbb{E}X\|^2 = \mathbb{E}\left[\|X\|^2\right] - \|\mathbb{E}[X]\|^2 \leq \mathbb{E}\left[\|X\|^2\right]$, (c) follows from Assumption 1 (which implies that $|L(\mathbf{T})| \leq C$ for all $\mathbf{T}$ and also consequently that $|L_{avg}| \leq C$), and (d) follows from the Cauchy-Schwarz inequality. Therefore, for any $i \in [n]$, $\text{Var}\left(\boldsymbol{g}_{\boldsymbol{p}_i}\right)$ is indeed bounded, and consequently, the final gradient estimator is also bounded. $\qquad\square$

## B.4 Proofs for Theorem 1

**Lemma 3.** *Let $\{\boldsymbol{p}\}_{t=1}^T$ be generated by mDP-DPG. Let $\eta_t \in (0,1]$ and $\gamma \in (0, \frac{1}{2L\eta_t}]$, we have*

$$\mathcal{L}(\boldsymbol{p}_{t+1}) \leq \mathcal{L}(\boldsymbol{p}_t) + \eta_t\gamma\|\nabla\mathcal{L}(\boldsymbol{p}_t) - \boldsymbol{m}_t\|^2 - \frac{\eta_t}{2\gamma}\|\tilde{\boldsymbol{p}}_{t+1} - \boldsymbol{p}_t\|^2. \tag{16}$$

*Proof.* Recall that $\mathcal{L}(\boldsymbol{p}) := \mathbb{E}_S \mathbb{E}_{\mathbf{T} \sim \mathcal{P}_{\boldsymbol{p}}} L_B(h([\mathbf{T}, \cdot]), S)$. According to Lemma 2, $\mathcal{L}(\boldsymbol{p})$ is $L$-smooth. Then we have

$$
\begin{aligned}
\mathcal{L}(\boldsymbol{p}_{t+1}) \leq & \mathcal{L}(\boldsymbol{p}_t) + \langle \nabla \mathcal{L}(\boldsymbol{p}_t), \boldsymbol{p}_{t+1} - \boldsymbol{p}_t \rangle + \frac{L}{2} \|\boldsymbol{p}_{t+1} - \boldsymbol{p}_t\|^2 \\
\leq & \mathcal{L}(\boldsymbol{p}_t) + \eta_t \langle \nabla \mathcal{L}(\boldsymbol{p}_t), \tilde{\boldsymbol{p}}_{t+1} - \boldsymbol{p}_t \rangle + \frac{L\eta_t^2}{2} \|\tilde{\boldsymbol{p}}_{t+1} - \boldsymbol{p}_t\|^2 \\
\leq & \mathcal{L}(\boldsymbol{p}_t) + \eta_t \langle \nabla \mathcal{L}(\boldsymbol{p}_t) - \boldsymbol{m}_t, \tilde{\boldsymbol{p}}_{t+1} - \boldsymbol{p}_t \rangle + \eta_t \langle \boldsymbol{m}_t, \tilde{\boldsymbol{p}}_{t+1} - \boldsymbol{p}_t \rangle + \frac{L\eta_t^2}{2} \|\tilde{\boldsymbol{p}}_{t+1} - \boldsymbol{p}_t\|^2.
\end{aligned}
$$

Since $\tilde{\boldsymbol{p}}_{t+1} = proj_{\mathcal{C}}(\boldsymbol{p}_t - \gamma \boldsymbol{m}_t) = \arg\min_{\boldsymbol{p} \in \mathcal{C}} \frac{1}{2} \|\boldsymbol{p} - (\boldsymbol{p}_t - \gamma \boldsymbol{m}_t)\|^2$, we have $\forall \boldsymbol{p} \in \mathcal{C}$, $\langle \tilde{\boldsymbol{p}}_{t+1} - (\boldsymbol{p}_t - \gamma \boldsymbol{m}_t), \boldsymbol{p} - \tilde{\boldsymbol{p}}_{t+1} \rangle \geq 0$. Set $\boldsymbol{p} = \boldsymbol{p}_t$, we have

$$
\langle \boldsymbol{m}_t, \tilde{\boldsymbol{p}}_{t+1} - \boldsymbol{p}_t \rangle \leq -\frac{1}{\gamma} \|\tilde{\boldsymbol{p}}_{t+1} - \boldsymbol{p}_t\|^2.
$$

Thus we have

$$
\begin{aligned}
& \mathcal{L}(\boldsymbol{p}_{t+1}) \\
\leq & \mathcal{L}(\boldsymbol{p}_t) + \eta_t \langle \nabla \mathcal{L}(\boldsymbol{p}_t) - \boldsymbol{m}_t, \tilde{\boldsymbol{p}}_{t+1} - \boldsymbol{p}_t \rangle + \eta_t \langle \boldsymbol{m}_t, \tilde{\boldsymbol{p}}_{t+1} - \boldsymbol{p}_t \rangle + \frac{L\eta_t^2}{2} \|\tilde{\boldsymbol{p}}_{t+1} - \boldsymbol{p}_t\|^2 \\
\leq & \mathcal{L}(\boldsymbol{p}_t) + \eta_t \gamma \|\nabla \mathcal{L}(\boldsymbol{p}_t) - \boldsymbol{m}_t\|^2 + \frac{\eta_t}{4\gamma} \|\tilde{\boldsymbol{p}}_{t+1} - \boldsymbol{p}_t\|^2 - \frac{\eta_t}{\gamma} \|\tilde{\boldsymbol{p}}_{t+1} - \boldsymbol{p}_t\|^2 + \frac{L\eta_t^2}{2} \|\tilde{\boldsymbol{p}}_{t+1} - \boldsymbol{p}_t\|^2 \\
= & \mathcal{L}(\boldsymbol{p}_t) + \eta_t \gamma \|\nabla \mathcal{L}(\boldsymbol{p}_t) - \boldsymbol{m}_t\|^2 - \frac{\eta_t}{2\gamma} \|\tilde{\boldsymbol{p}}_{t+1} - \boldsymbol{p}_t\|^2 - (\frac{\eta_t}{4\gamma} - \frac{L\eta_t^2}{2}) \|\tilde{\boldsymbol{p}}_{t+1} - \boldsymbol{p}_t\|^2 \\
\leq & \mathcal{L}(\boldsymbol{p}_t) + \eta_t \gamma \|\nabla \mathcal{L}(\boldsymbol{p}_t) - \boldsymbol{m}_t\|^2 - \frac{\eta_t}{2\gamma} \|\tilde{\boldsymbol{p}}_{t+1} - \boldsymbol{p}_t\|^2.
\end{aligned}
$$

Where the last inequality holds due to $0 < \gamma < \frac{1}{2L\eta_t}$.

$\square$

**Lemma 4.**

$$
\mathbb{E}\|\nabla \mathcal{L}(\boldsymbol{p}_{t+1}) - \boldsymbol{m}_{t+1}\|^2 \tag{17}
$$
$$
\leq (1-\theta_t)^2 \mathbb{E}\|\nabla \mathcal{L}(\boldsymbol{p}_t) - \boldsymbol{m}_t\|^2 + 2(1-\theta_t)^2 L^2 \eta_t^2 \|\tilde{\boldsymbol{p}}_{t+1} - \boldsymbol{p}_t\|^2 + 2\theta_t^2 \sigma^2. \tag{18}
$$

*Proof.* According to the update rule of $\boldsymbol{m}_{t+1}$, we have

$$
\begin{aligned}
& \mathbb{E}\|\nabla \mathcal{L}(\boldsymbol{p}_{t+1}) - \boldsymbol{m}_{t+1}\|^2 \\
= & \mathbb{E}\|\nabla \mathcal{L}(\boldsymbol{p}_{t+1}) - \boldsymbol{g}_{\boldsymbol{p}_{t+1},S_{t+1}} - (1-\theta_{t+1})(\boldsymbol{m}_t - \boldsymbol{g}_{\boldsymbol{p}_t,S_{t+1}})\|^2 \\
= & \mathbb{E}\|(1-\theta_{t+1})(\nabla \mathcal{L}(\boldsymbol{p}_t) - \boldsymbol{m}_t) + \theta_t(\nabla \mathcal{L}(\boldsymbol{p}_{t+1}) - \boldsymbol{g}_{\boldsymbol{p}_{t+1},S_{t+1}}) \\
& + (1-\theta_{t+1})(\nabla \mathcal{L}(\boldsymbol{p}_{t+1}) - \nabla \mathcal{L}(\boldsymbol{p}_t) - (\boldsymbol{g}_{\boldsymbol{p}_{t+1},S_{t+1}} - \boldsymbol{g}_{\boldsymbol{p}_t,S_{t+1}}))\|^2 \\
\leq & (1-\theta_t)^2 \mathbb{E}\|\nabla \mathcal{L}(\boldsymbol{p}_t) - \boldsymbol{m}_t\|^2 + 2(1-\theta_t)^2 \|\nabla \mathcal{L}(\boldsymbol{p}_{t+1}) - \nabla \mathcal{L}(\boldsymbol{p}_t) - (\boldsymbol{g}_{\boldsymbol{p}_{t+1},S_{t+1}} - \boldsymbol{g}_{\boldsymbol{p}_t,S_{t+1}})\|^2 \\
& + 2\theta_t^2 \mathbb{E}\|\nabla \mathcal{L}(\boldsymbol{p}_{t+1}) - \boldsymbol{g}_{\boldsymbol{p}_{t+1},S_{t+1}}\|^2 \\
\leq & (1-\theta_t)^2 \mathbb{E}\|\nabla \mathcal{L}(\boldsymbol{p}_t) - \boldsymbol{m}_t\|^2 + 2(1-\theta_t)^2 \|\boldsymbol{g}_{\boldsymbol{p}_{t+1},S_{t+1}} - \boldsymbol{g}_{\boldsymbol{p}_t,S_{t+1}}\|^2 + 2\theta_t^2 \sigma^2 \\
\leq & (1-\theta_t)^2 \mathbb{E}\|\nabla \mathcal{L}(\boldsymbol{p}_t) - \boldsymbol{m}_t\|^2 + 2(1-\theta_t)^2 L^2 \|\boldsymbol{p}_{t+1} - \boldsymbol{p}_t\|^2 + 2\theta_t^2 \sigma^2 \\
= & (1-\theta_t)^2 \mathbb{E}\|\nabla \mathcal{L}(\boldsymbol{p}_t) - \boldsymbol{m}_t\|^2 + 2(1-\theta_t)^2 L^2 \eta_t^2 \|\tilde{\boldsymbol{p}}_{t+1} - \boldsymbol{p}_t\|^2 + 2\theta_t^2 \sigma^2.
\end{aligned}
$$

$\square$

*Proof of Theorem.*

$$\frac{1}{\eta_t}\mathbb{E}\|\nabla\mathcal{L}(\boldsymbol{p}_{t+1}) - \boldsymbol{m}_{t+1}\|^2 - \frac{1}{\eta_{t-1}}\mathbb{E}\|\nabla\mathcal{L}(\boldsymbol{p}_t) - \boldsymbol{m}_t\|^2$$

$$\leq(\frac{(1-\theta_t)^2}{\eta_t} - \frac{1}{\eta_{t-1}})\mathbb{E}\|\nabla\mathcal{L}(\boldsymbol{p}_t) - \boldsymbol{m}_t\|^2 + 2(1-\theta_t)^2 L^2\eta_t\|\tilde{\boldsymbol{p}}_{t+1} - \boldsymbol{p}_t\|^2 + \frac{2\theta_t^2\sigma^2}{\eta_t}$$

$$\leq(\frac{1-\theta_t}{\eta_t} - \frac{1}{\eta_{t-1}})\mathbb{E}\|\nabla\mathcal{L}(\boldsymbol{p}_t) - \boldsymbol{m}_t\|^2 + 2L^2\eta_t\|\tilde{\boldsymbol{p}}_{t+1} - \boldsymbol{p}_t\|^2 + \frac{2\theta_t^2\sigma^2}{\eta_t}$$

$$=(\frac{1}{\eta_t} - \frac{1}{\eta_{t-1}} - c\eta_t)\mathbb{E}\|\nabla\mathcal{L}(\boldsymbol{p}_t) - \boldsymbol{m}_t\|^2 + 2L^2\eta_t\|\tilde{\boldsymbol{p}}_{t+1} - \boldsymbol{p}_t\|^2 + \frac{2\theta_t^2\sigma^2}{\eta_t}.$$

Let $\eta_t = \frac{k}{(\xi+t)^{1/3}}$, we have

$$\frac{1}{\eta_t} - \frac{1}{\eta_{t-1}} =\frac{1}{k}((\xi+t)^{1/3} - (\xi+t-1)^{1/3}) \leq \frac{1}{3k(\xi+t-1)^{2/3}} \leq \frac{1}{3k(\xi/2+t)^{2/3}}$$

$$\leq\frac{2^{2/3}}{3k(\xi+t)^{2/3}} = \frac{2^{2/3}}{3k^3}\eta_t^2 \leq \frac{2}{3k^3}\eta_t,$$

where the first inequality is due to $(x+1)^{1/3} \leq x^{1/3} + \frac{1}{3x^{2/3}}$ and the second inequality is due to $\xi > 2$. Let $c \geq \frac{2}{3k^3} + \frac{5}{4}$, then we have

$$\frac{1}{\eta_t}\mathbb{E}\|\nabla\mathcal{L}(\boldsymbol{p}_{t+1}) - \boldsymbol{m}_{t+1}\|^2 - \frac{1}{\eta_{t-1}}\mathbb{E}\|\nabla\mathcal{L}(\boldsymbol{p}_t) - \boldsymbol{m}_t\|^2$$

$$\leq -\frac{5}{4}\eta_t\mathbb{E}\|\nabla\mathcal{L}(\boldsymbol{p}_t) - \boldsymbol{m}_t\|^2 + 2L^2\eta_t\|\tilde{\boldsymbol{p}}_{t+1} - \boldsymbol{p}_t\|^2 + \frac{2\theta_t^2\sigma^2}{\eta_t}.$$

Then we define the *Lyapunov* function $R_t = \mathbb{E}[\mathcal{L}(\boldsymbol{p}_t) + \frac{\gamma}{\eta_{t-1}}\|\nabla\mathcal{L}(\boldsymbol{p}_t) - \boldsymbol{m}_t\|^2]$. Then we have

$$R_{t+1} - R_t =\mathbb{E}[\mathcal{L}(\boldsymbol{p}_{t+1}) - \mathcal{L}(\boldsymbol{p}_t)] + \frac{\gamma}{\eta_t}\mathbb{E}\|\nabla\mathcal{L}(\boldsymbol{p}_{t+1}) - \boldsymbol{m}_{t+1}\|^2 - \frac{\gamma}{\eta_{t-1}}\mathbb{E}\|\nabla\mathcal{L}(\boldsymbol{p}_t) - \boldsymbol{m}_t\|^2$$

$$\leq(\eta_t\gamma - \frac{5\eta_t\gamma}{4})\mathbb{E}\|\nabla\mathcal{L}(\boldsymbol{p}_t) - \boldsymbol{m}_t\|^2 - \frac{\eta_t}{2\gamma}\mathbb{E}\|\tilde{\boldsymbol{p}}_{t+1} - \boldsymbol{p}_t\|^2 + 2L^2\eta_t\gamma\|\tilde{\boldsymbol{p}}_{t+1} - \boldsymbol{p}_t\|^2 + \frac{2\theta_t^2\sigma^2\gamma}{\eta_t}$$

$$\leq -\frac{\eta_t\gamma}{4}\mathbb{E}\|\nabla\mathcal{L}(\boldsymbol{p}_t) - \boldsymbol{m}_t\|^2 - \frac{\eta_t}{4\gamma}\mathbb{E}\|\tilde{\boldsymbol{p}}_{t+1} - \boldsymbol{p}_t\|^2 + \frac{2\theta_t^2\sigma^2\gamma}{\eta_t},$$

where the last inequality is due to $\gamma \leq \frac{1}{2\sqrt{2}L}$. Rearranging the above inequality, we have

$$\frac{\eta_t\gamma}{4}\mathbb{E}\|\nabla\mathcal{L}(\boldsymbol{p}_t) - \boldsymbol{m}_t\|^2 + \frac{\eta_t}{4\gamma}\mathbb{E}\|\tilde{\boldsymbol{p}}_{t+1} - \boldsymbol{p}_t\|^2 \leq R_t - R_{t+1} + \frac{2\theta_t^2\sigma^2\gamma}{\eta_t}.$$

Taking average over timesteps $t = 1, \ldots, T$, we have

$$\frac{1}{T}\sum_{t=1}^{T}\mathbb{E}\left[\frac{\eta_t\gamma}{4}\|\nabla\mathcal{L}(\boldsymbol{p}_t) - \boldsymbol{m}_t\|^2 + \frac{\eta_t}{4\gamma}\|\tilde{\boldsymbol{p}}_{t+1} - \boldsymbol{p}_t\|^2\right]$$

$$\leq\frac{\mathcal{L}(\boldsymbol{p}_1) - \mathcal{L}^*}{T} + \frac{\gamma\|\nabla\mathcal{L}(\boldsymbol{p}_1) - \boldsymbol{m}_1\|^2}{T\eta_0} + \sum_{t=1}^{T}\frac{2\theta_t^2\sigma^2\gamma}{T\eta_t} \leq \frac{\mathcal{L}(\boldsymbol{p}_1) - \mathcal{L}^*}{T} + \frac{\gamma\sigma^2}{T\eta_0} + \sum_{t=1}^{T}\frac{2\theta_t^2\sigma^2\gamma}{T\eta_t}$$

$$=\frac{\mathcal{L}(\boldsymbol{p}_1) - \mathcal{L}^*}{T} + \frac{\gamma\xi^{1/3}\sigma^2}{kT} + \sum_{t=1}^{T}\frac{2c^2\eta_t^3\sigma^2\gamma}{T}.$$

Dividing both sides with $\gamma\eta_T$, we have

$$\frac{1}{T}\sum_{t=1}^{T}\mathbb{E}\left[\frac{1}{4}\|\nabla\mathcal{L}(\boldsymbol{p}_t)-\boldsymbol{m}_t\|^2+\frac{1}{4\gamma^2}\|\tilde{\boldsymbol{p}}_{t+1}-\boldsymbol{p}_t\|^2\right]$$

$$\leq\frac{\mathcal{L}(\boldsymbol{p}_1)-\mathcal{L}^*}{T\eta_T\gamma}+\frac{\xi^{1/3}\sigma^2}{kT\eta_T}+\sum_{t=1}^{T}\frac{2c^2\eta_t^3\sigma^2}{T\eta_T}\leq\frac{\mathcal{L}(\boldsymbol{p}_1)-\mathcal{L}^*}{T\eta_T\gamma}+\frac{\xi^{1/3}\sigma^2}{kT\eta_T}+\frac{2c^2\sigma^2}{T\eta_T}\int_{1}^{T}\frac{k^3}{\xi+t}dt$$

$$\leq\frac{\mathcal{L}(\boldsymbol{p}_1)-\mathcal{L}^*}{T\eta_T\gamma}+\frac{\xi^{1/3}\sigma^2}{kT\eta_T}+\frac{2c^2\sigma^2k^3}{T\eta_T}\log(\xi+T)$$

$$=\frac{\mathcal{L}(\boldsymbol{p}_1)-\mathcal{L}^*}{T\gamma k}(\xi+T)^{\frac{1}{3}}+\frac{\xi^{1/3}\sigma^2}{kT}(\xi+T)^{\frac{1}{3}}+\frac{2c^2\sigma^2k^3}{T}(\xi+T)^{\frac{1}{3}}\log(\xi+T)$$

$$\leq\frac{M}{T}(\xi+T)^{\frac{1}{3}},$$

where $M=\frac{\mathcal{L}(\boldsymbol{p}_1)-\mathcal{L}^*}{\gamma k}+\frac{\xi^{1/3}\sigma^2}{k}+2c^2\sigma^2k^3\log(\xi+T)$. Using Jensen's inequality, we have

$$\frac{1}{T}\sum_{t=1}^{T}\mathbb{E}\left[\frac{1}{2}\|\nabla\mathcal{L}(\boldsymbol{p}_t)-\boldsymbol{m}_t\|+\frac{1}{2\gamma}\|\tilde{\boldsymbol{p}}_{t+1}-\boldsymbol{p}_t\|\right]$$

$$\leq\left(\frac{2}{T}\sum_{t=1}^{T}\mathbb{E}\left[\frac{1}{4}\|\nabla\mathcal{L}(\boldsymbol{p}_t)-\boldsymbol{m}_t\|^2+\frac{1}{4\gamma^2}\|\tilde{\boldsymbol{p}}_{t+1}-\boldsymbol{p}_t\|^2\right]\right)^{\frac{1}{2}}$$

$$\leq\frac{\sqrt{2M}}{T^{1/2}}(\xi+T)^{1/6}\leq\frac{\sqrt{2M}}{T^{1/2}}(\xi^{1/6}+T^{1/6}),$$

where the first inequality is due to $x+y\leq(2x^2+2y^2)^{1/2}$ and the last inequality is due to $(x+y)^{1/6}\leq x^{1/6}+y^{1/6}$. Then we have

$$\frac{1}{T}\sum_{t=1}^{T}\mathbb{E}\|G_{\mathcal{C}}(\boldsymbol{p}_t,\gamma)\|=\frac{1}{T}\sum_{t=1}^{T}\frac{1}{\gamma}\mathbb{E}\|\boldsymbol{p}_t-\text{proj}_{\mathcal{C}}(\boldsymbol{p}_t-\gamma\nabla\mathcal{L}(\boldsymbol{p}_t))\|$$

$$=\frac{1}{T}\sum_{t=1}^{T}\frac{1}{\gamma}\mathbb{E}\|\boldsymbol{p}_t-\text{proj}_{\mathcal{C}}(\boldsymbol{p}_t-\gamma\boldsymbol{m}_t)+\text{proj}_{\mathcal{C}}(\boldsymbol{p}_t-\gamma\boldsymbol{m}_t)-\text{proj}_{\mathcal{C}}(\boldsymbol{p}_t-\gamma\nabla\mathcal{L}(\boldsymbol{p}_t))\|$$

$$=\frac{1}{T}\sum_{t=1}^{T}\frac{1}{\gamma}\mathbb{E}\|\boldsymbol{p}_t-\tilde{\boldsymbol{p}}_{t+1}+\text{proj}_{\mathcal{C}}(\boldsymbol{p}_t-\gamma\boldsymbol{m}_t)-\text{proj}_{\mathcal{C}}(\boldsymbol{p}_t-\gamma\nabla\mathcal{L}(\boldsymbol{p}_t))\|$$

$$\leq\frac{1}{T}\sum_{t=1}^{T}\frac{1}{\gamma}\mathbb{E}\left[\|\boldsymbol{p}_t-\tilde{\boldsymbol{p}}_{t+1}\|+\|\text{proj}_{\mathcal{C}}(\boldsymbol{p}_t-\gamma\boldsymbol{m}_t)-\text{proj}_{\mathcal{C}}(\boldsymbol{p}_t-\gamma\nabla\mathcal{L}(\boldsymbol{p}_t))\|\right]$$

$$\leq\frac{1}{T}\sum_{t=1}^{T}\mathbb{E}\left[\|\nabla\mathcal{L}(\boldsymbol{p}_t)-\boldsymbol{m}_t\|+\frac{1}{\gamma}\|\boldsymbol{p}_t-\tilde{\boldsymbol{p}}_{t+1}\|\right]$$

$$\leq\frac{2\sqrt{2M}}{T^{1/2}}\xi^{1/6}+\frac{2\sqrt{2M}}{T^{1/3}},$$

where the first inequality is due to Triangle inequality, the second inequality is due to the non-expansivity of convex projection.

$\square$

## C    IMPLEMENTATION DETAILS

### C.1    MANUAL TEMPLATES

Table 4: Input templates, and output label words used in RoBERTa-large. $\langle S \rangle$ represents the sentences in the dataset. $[MASK]$ represents the mask token.

| Task | Dataset | Input Template | Output Label Words |
|------|---------|----------------|--------------------|
| Real-World Datasets | Amazon Book | $\langle S \rangle$ It was $[MASK]$. | positive, negative |
|                     | Amazon Electronics | $\langle S \rangle$ It was $[MASK]$. | positive, negative |
| GLUE Datasets | CoLA | $\langle S \rangle$ correct? $[MASK]$. | no, yes |
|               | MRPC | $\langle S_1 \rangle \langle S_2 \rangle$ entailment? $[MASK]$. | no, yes |

Table 5: Input templates, and output label words used in GPT2-XL and Llama3. $\langle S \rangle$ represents the sentences in the dataset.

| Task | Dataset | Input Template | Output Label Words |
|------|---------|----------------|--------------------|
| Real-World Datasets | Amazon Book | $\langle S \rangle$ It was | positive, negative |
|                     | Amazon Electronics | $\langle S \rangle$ It was | positive, negative |
| GLUE Datasets | CoLA | $\langle S \rangle$ correct? | no, yes |
|               | MRPC | $\langle S_1 \rangle \langle S_2 \rangle$ entailment? | no, yes |

### C.2    HYPERPARAMETERS

Table 6: Main hyperparameters used in our algorithm.

| Hyperparameter | RoBERTa-large | GPT2-XL | Llama3 |
|----------------|---------------|---------|--------|
| query limit | 32000 | 3200 | 1600 |
| train batch size | 32 | 32 | 16 |
| prompt length | $\{50, 20\}$ | $\{50, 20\}$ | $\{50, 20\}$ |
| step size | 1e-3 | 1e-3 | 1e-3 |

## D    ADDITIONAL EXPERIMENT RESULTS

### D.1    EXPERIMENT RESULTS ON MRPC

We conduct experiments on MRPC using the same experiment setups as on CoLA and observe similar phenomena as those on CoLA. The results are shown in Table 7.

Table 7: Comparison of AUC scores (mean±std.) on constructed imbalanced scenarios of MRPC. We conduct three groups of experiments on pre-trained RoBERTa-large, GPT2-XL, and Llama3 with a prompt length of 20. The best results are highlighted in **bold**.

| Imbalanced Ratio | Method | RoBERTa-large | GPT2-XL | Llama3 |
|------------------|--------|---------------|---------|--------|
| $\tau = 20$ | Manual Prompt | .4764±.0855 | .4556±.0834 | **.5264±.0531** |
|             | BBT | .5236±.0497 | .4986±.0024 | .5000±.0000 |
|             | GAP3 | .5000±.0000 | .4972±.0024 | .4708±.0331 |
|             | BDPL | .4639±.1660 | .4917±.0072 | .4972±.0048 |
|             | mDP-DPG (ours) | **.5292±.0573** | **.5278±.0808** | .5083±.0144 |
| $\tau = 50$ | Manual Prompt | .4700±.1386 | .4433±.1444 | .5206±.1275 |
|             | BBT | .5400±.0173 | .4972±.0024 | .5250±.0433 |
|             | GAP3 | .5000±.0000 | .5517±.1721 | .4717±.0407 |
|             | BDPL | .3050±.0976 | .5667±.1241 | .5000±.0000 |
|             | mDP-DPG (ours) | **.5767±.0580** | **.5983±.1234** | **.5317±.0548** |

## D.2 Overcoming the Limitation of Binary Classification

Although our current use of AUC loss is specific to binary classification, however, it could also be generalized to the multi-class classification dataset by using the micro averaging. That is, for each class, calculate the AUC loss for that class against all others and sum the losses and average them over all classes. Additionally, we conduct experiments on the multi-class datasets MNLI and SNLI and compare our methods with 3 representative baselines. The results are presented in the Table 8. It can be observed that our methods also maintain optimal performance on multi-class datasets.

Table 8: Comparison of AUC values on 2 multi-class datasets

| Model | Method | MNLI (len=50) | SNLI (len=50) |
|---|---|---|---|
| RoBERTa-Large | Manual Prompt | .4636±.0340 | .5290±.0458 |
| | BBT | .4589±.0271 | .5845±.0470 |
| | BDPL | .4670±.0462 | .5697±.0189 |
| | mDP-DPG(ours) | **.4814±.0538** | **.5897±.0296** |
| GPT2-XL | Manual Prompt | .4718±.0360 | .5150±.0434 |
| | BBT | .4761±.0413 | .5177±.0113 |
| | BDPL | .4518±.0503 | .5104±.0189 |
| | mDP-DPG(ours) | **.4823±.0382** | **.5243±.0197** |
| Llama3 | Manual Prompt | .4036±.0106 | .5478±.0122 |
| | BBT | .5025±.0034 | .4612±.0556 |
| | BDPL | .4946±.0124 | .6034±.0233 |
| | mDP-DPG(ours) | **.5079±.0284** | **.6171±.0298** |

## D.3 Balanced Scenario

Although experiments in the paper have demonstrated that our methods outperform baseline methods in imbalanced scenarios, their effectiveness in balanced settings is equally important. If our methods were to suffer from performance collapse in balanced scenarios, their utility would be compromised. To verify the performance of our methods in the 16-shot setting, we have conducted additional experiments, with the results provided in Table 9. In balanced scenarios, the performance of our methods is similar to that of various baselines in most cases, with a few instances where they even surpass all baselines.

Table 9: Comparison of ACC on 3 datasets in the balanced scenario with prompt length of 20.

| Model | Method | CoLA | Book | Elec |
|---|---|---|---|---|
| RoBERTa-Large | BBT | .5717±.0159 | .9364±.0008 | **.9149±.0040** |
| | GAP3 | .5254±.0739 | .9019±.0308 | .8519±.1271 |
| | BDPL | .4851±.0515 | .9349±.0016 | .8794±.0229 |
| | mDP-DPG(ours) | **.5762±.0605** | **.9384±.0006** | .9112±.0075 |
| GPT2-XL | BBT | .3321±.0244 | .6873±.0397 | .5423±.0671 |
| | GAP3 | .4624±.0774 | **.8257±.1216** | .6312±.3736 |
| | BDPL | **.6497±.0221** | .7527±.0283 | .6741±.0941 |
| | mDP-DPG(ours) | .6142±.0339 | .8034±.0217 | **.7777±.0608** |

## D.4 Training Efficiency

On the one hand, both mDP-DPG and BDPL use variance-reduced policy gradient estimators. On the other hand, since mDP-DPG requires sampling pairs of examples and involves additional forward passes through the black-box model. To mitigate this computational cost, we employ smaller mini-batch size, which reduces the number of forward passes while achieving comparable experimental results. Furthermore, although pairwise sampling necessitates multiple forward passes, the computational burden is still much lower when compared backpropagation. To visually compare the training efficiency between BDPL and our methods, we provide Figure 4 showing the progression of the current best AUC on the development set across epochs.

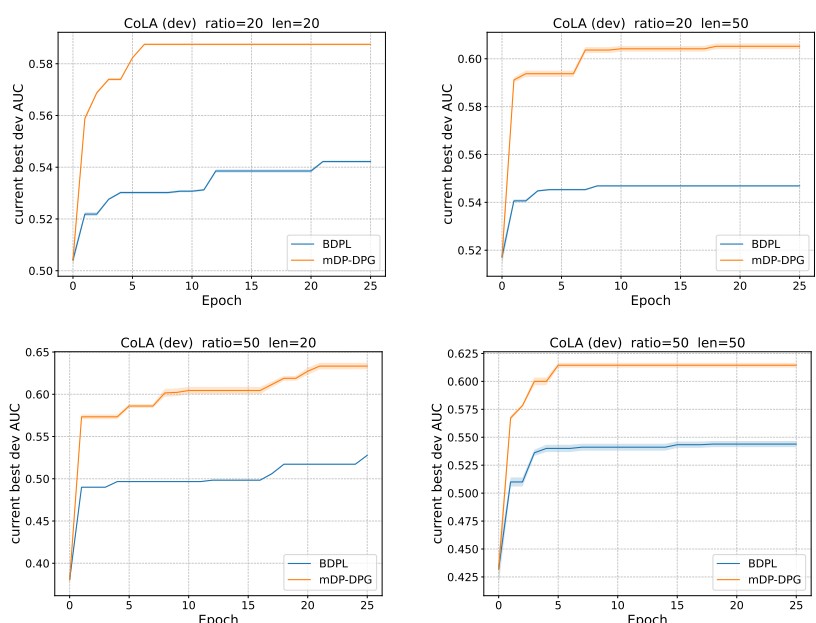

Figure 4: Current best AUC on the development set.

### D.5 LEARNED PROMPT AND INTERPRETABILITY

We provide the prompts learned by our methods in Table 10 and 11, along with some correctly predicted examples. Our prompts, like those in Diao et al. (2022), are sequences of discrete words without explicit natural language semantics. Additionally, from the black-box optimization perspective, we prefer to consider the prompts as tunable parameters of LLM, and we can adapt the model to downstream tasks at a lower cost by optimizing prompts.

### D.6 REAL-WORLD APPLIACTION

To show performance of our method in real-world applications, we add three additional representative imbalance datasets: **BB (Burfoot and Baldwin)** Burfoot & Baldwin (2009), originally developed for satire news detection; **Job Scams** and **SMS** datasets Boumber et al. (2024), derived for fraudulent job postings and spam message detection, respectively. The BB dataset consists of 4,000 true news articles and 233 satire articles. Its challenge lies in satire articles mimicking the tone and style of true news while incorporating exaggerated or absurd content, requiring semantic understanding and background knowledge for accurate classification. The Job Scams dataset, derived from the Employment Scam Aegean Dataset, includes 14,295 cleaned job advertisements, of which 599 are fraudulent postings, presenting a significant class imbalance. Fraudulent postings often use fake job positions to deceive applicants, typically featuring short and structured texts. The SMS dataset contains 6,574 messages, of which 1,274 are spam or phishing. These deceptive messages are typically brief and generic promotional content, whereas genuine messages reflect more personalized communication. The diversity of these datasets ensures the broad applicability of the experiments.

We compare the performance of **five methods—Manual Prompt**, **BDPL**, **APE** (Zhou et al., 2022), **EvoPrompt** (Guo et al., 2023), and **mDP-DPG**—evaluated under prompt lengths of 5 and 20 using true black-box LLM GPT-4. For APE and EvoPrompt, the final prompts are generated based on the manual prompt pool and therefore do not have a fixed prompt length. The results are shown in Table 12. The experimental results demonstrate that mDP-DPG outperforms other methods across all datasets. On the BB dataset, mDP-DPG achieves an AUC of 0.5972 (length = 5), significantly surpassing BDPL and Manual Prompt. Despite the class imbalance challenge on the Job Scams dataset, mDP-DPG achieves an AUC of 0.5307 (length = 20), outperforming BDPL's 0.5024. Overall, the re-

Table 10: Learned prompts on RoBERTa-Large and correctly classified examples

| Method | Dataset | Prompt+Sentence | Prediction | Label |
|---|---|---|---|---|
| mDP-DPG | CoLA | Sandy was trying to work out which students would be able to solve a certain problem, but she wouldn't tell us which one. | no | yes |
| | | This never in had Tom of her It his if They than with some not know think would That my Sandy was trying to work out which students would be able to solve a certain problem, but she wouldn't tell us which one. | yes | |
| | Book | Not impressed! This is just another gluten free cookbook - albeit with some great recipes. I don't find recipes that contain agave nectar to be "sugar" free. What's the purpose - anyone can use other sweeteners than plain old sugar... While the recipes might be sugar-free, they are definitely NOT without sweeteners. Sorry I bought the book...perhaps there are better books using NO sweeteners that satisfy the sweet tooth of a gluten-free person??? | positive | negative |
| | | is their books at is one It through good can really he I so will just It never so will Not impressed! This is just another gluten free cookbook - albeit with some great recipes. I don't find recipes that contain agave nectar to be "sugar" free. What's the purpose - anyone can use other sweeteners than plain old sugar... While the recipes might be sugar-free, they are definitely NOT without sweeteners. Sorry I bought the book...perhaps there are better books using NO sweeteners that satisfy the sweet tooth of a gluten-free person??? | negative | |

Table 11: Learned prompts on GPT2-XL and correctly classified examples

| Method | Dataset | Prompt+Sentence | Prediction | Label |
|---|---|---|---|---|
| mDP-DPG | CoLA | Jerry attempted to blow up the Pentagon. | no | yes |
| | | which did not by her from people What him do eat were people will on It a more was made Jerry attempted to blow up the Pentagon. | yes | |
| | Book | I had read the 1 year to an organized life, by Ms Leeds, I thought this was going to be different. It's the exact same book!! I recommend buying the newer version: 1 year to an organized life, don't waste money on this one. | positive | negative |
| | | well into there most love reading into a most characters not up she their a my book people only most I had read the 1 year to an organized life, by Ms Leeds, I thought this was going to be different. It's the exact same book!! I recommend buying the newer version: 1 year to an organized life, don't waste money on this one. | negative | |

Table 12: Comparison of AUC values across different datasets using GPT-4.

| Length | Method | BB ($\tau = 20$) | Job Scams ($\tau = 20$) | SMS ($\tau = 5$) |
|--------|--------|------------------|-------------------------|------------------|
| - | Manual Prompt | .4333±.0191 | .5098±.0546 | .5252±.0131 |
| ≤20 | APE | .4968±.0398 | .5000±.0042 | .5268±.0184 |
| ≤20 | EvoPrompt | .5002±.0373 | .4972±.0064 | .5271±.0231 |
| 5 | BDPL | .4486±.0146 | .4987±.0235 | .5200±.0114 |
| | mDP-DPG (ours) | **.5972±.1428** | **.5314±.0386** | **.5272±.0090** |
| 20 | BDPL | .4403±.0244 | .5024±.0306 | .5135±.0064 |
| | mDP-DPG (ours) | **.5236±.1545** | **.5307±.0043** | **.5278±.0119** |

sults validate the adaptability of the mDP-DPG method, particularly in complex or class-imbalanced tasks where it exhibited remarkable advantages.

