# OpenReview forum: "Dual Variance Reduction with Momentum for Imbalanced Black-Box Discrete Prompt Learning"
_ICLR.cc/2025/Conference — Submitted to ICLR 2025_

### Official Review · Reviewer_pkzH · 2024-11-01

**Soundness:** 4
**Presentation:** 3
**Contribution:** 3
**Rating:** 6
**Confidence:** 3

**Summary:**

This paper tackles the challenges of learning discrete prompts for large language models (LLMs) in the context of imbalanced data distributions. The authors integrate pairwise AUC loss into a policy gradient optimization framework and introduce a doubly policy gradient approach. To improve stability, they introduce the variance-reduced doubly policy gradient estimator (VR-DPGE) and the STORM technique into their method, momentum-based discrete prompt learning with doubly policy gradient (mDP-DPG). They demonstrate their method with theoretical proof and experimental results.

**Strengths:**

1. the paper presents a novel approach for optimizing prompts in the context of imbalanced classification problems with LLMs.  The authors introduce VR-DPOGE and STROM strategy to  effectively address the dual variance involved in gradient calculation.
2. The paper offers rigorous theoretical guarantees for the mDP-DPG method. Additionally, the experimental results validate the effectiveness of the approach, demonstrating a clear advancement over baselines across diverse imbalanced text classification datasets.

**Weaknesses:**

The paper lacks sufficient explanation on how it addresses the stated problem.

For instance, while the authors introduce the "unbiased VR-DPGE estimator," they do not provide adequate references for VR-DPGE, nor do they explain how Equation (8) specifically reduces variance. Some notation such as $T^{(k)}$  used in Equation (8) is not clearly defined.

A similar issue arises with the STORM strategy, where there is no explanation of the role of momentum. The equation appears only within the algorithm section. There is no definition on the  proj$_{C}$ function.

The theoretical results also require further discussion on what the theorem indicates about the resulting prompts.

**Questions:**

1. Please provide more intuitive discussion on how the VR-DPOGE and STORM strategy can reduce the variance of gradient estimation.

2. The paper assumes  independent categorical distribution on each token distribution, and proves that this distribution will converge. However, will the distribution converge to a delta distribution centered on a single optimal prompt? Could a correlated distribution over prompt tokens lead to convergence on such a single optimal prompt and potentially improving performance?

3. In Tables 1 and 2, several AUC values fall below 0.5. What could be the possible reasons for this performance being worse than a random guess?

---

> ### Author Response · Authors · 2024-11-24
> **Response to Reviewer pkzH**
>
> We appreciate the time and effort put into reviewing our manuscript. Please find our responses to your concerns below:
>
> **weakness & question 1:**
> Thanks for your valuable suggestion. In the revised version, we provide more discussion on how the VR-DPGE and STORM strategy can reduce the variance of gradient estimation and on what the theorem indicates about the resulting prompts. Then we define previously unclear notations. You can also find the relevant content below:
>
> * Discussion about VR-DPGE: We implement VR-DPGE, which incorporates a baseline subtraction term to reduce variance [1]. Compared to the high variance of loss values, the difference between the loss value and the baseline term has a lower variance, which facilitates more stable gradient estimation.
>
> * Discussion about STORM strategy: The content of Lemma 4 demonstrates that the momentum-based strategy effectively reduces the dual variance. The bound in Lemma 4 contains terms that decay with the momentum parameter $\theta_t$, showing that variance of the moving average $\boldsymbol{m}_{t+1}$ is systematically reduced over iterations, leading to a more stable and accurate approximation of the true gradient.
>
> * Discussion about theoretical results: The theorems demonstrate that the proposed framework, with variance-reduction techniques and momentum-based updates, ensures convergence towards a prompt distribution that minimizes the empirical risk of the pairwise AUC loss. This implies that the learned prompts are expected to be optimal in terms of performance for the downstream task under the given black-box constraints.
>
> * Definition of $T^{(k)}$: $T^{(k)}$ represents the $k$-th prompt obtained from $I$ prompt samplings.
>
> * Definition of $proj_{\mathcal{C}}$: $proj_{\mathcal{C}}$ in the update step is a projection function that projects updated prompt distribution to the constraint set $\mathcal{C}$ following [2].
>
> It is worth mentioning that the unbiased VR-DPGE estimator is first proposed in our paper, making it one of the key novelties of our work. As for the projection function $proj_{\mathcal{C}}$, we follow the approach in [2], and therefore is not elaborated upon in detail in the paper.
>
> **question 2:**
> Thank you for pointing out this concern. The assumption of independent sampling from the marginal distribution is a simplification that facilitates optimization in black-box settings, especially when the goal is to learn discrete prompts effectively. In fact, each probability distribution in the final solution does not ideally converge to a delta distribution. In order to obtain a stable prompt, following the approach in [2], the token at a given position in the final prompt is selected as the token with the highest probability in its corresponding probability distribution.
>
> **question 3:**
> Thanks for your question. For Manual Prompt, we attribute this phenomenon to the fact that Tables 1 and 2 in the original paper (Tables 1 and 7 in the revision) represent artificially constructed imbalanced scenarios, which lead to a deviation between the data distribution used in the experiments and the original distribution of the dataset. This deviation may cause the patterns learned by LLMs during pretraining to fail on the artificially constructed imbalanced data distribution. For BBT, BDPL, and GAP3, we believe the performance collapse of prompts learned on imbalanced data is due to the lack of techniques in these methods to handle imbalanced data, leading to the preference for majority class samples.
>
> [1] Greensmith et al., Variance reduction techniques for gradient estimates in reinforcement learning.
>
> [2] Diao et al., Black-box prompt learning for pre-trained language models.

---

> ### Author Response · Authors · 2024-12-02
>
> Dear Reviewer pkzH,
>
> Thank you once again for dedicating your time and effort to reviewing our manuscript. After our comprehensive replies and revisions, other reviewers have expressed increased support for our work and have improved their scores. If our responses meet your expectations, we hope you will consider offering further support for our work.
>
> Moreover, allow me to outline the additions we made during the rebuttal period:
>
> **Importance of the scenario:** We discuss the practical significance of classification tasks on imbalanced data and provide the example of recent advancement in the research community. For further details, please refer to our response to reviewer Ry7Y.
>
> **Additional experiments:** Based on the suggestion of reviewer Ry7Y and xsUR, we add experiments on real-world tasks using the **GPT-4** and compare our method with recent related works. Please refer to the Appendix D.6 in the revision for details.
>
> **Reiteration of contributions:** To aid in better understanding of the paper, we have reiterated the contributions, which are threefold: 1. Theoretical and technical improvements to address the limitations of the original framework. 2. A proposed approach for reducing dual variance. 3. A practical perspective (imbalanced data) on the application of prompt learning. For further details, please refer to our responses to reviewer Ry7Y and xsUR.
>
> **Justification of the assumption:** To address the concerns of reviewer 4JE3 and you, we explain the justification of the assumption that the token distributions are independent in our response to them.
>
> **Extension to multi-class problem:** We have previously addressed the extension to multi-class tasks in the original paper, which can be found in Appendix D.1 of original paper (Appendix D.2 of the revision).
>
> **Clarification of distribution convergence:** To address the concerns of reviewer 4JE3 and you, we clarify the distribution convergence effect in the experiments and explain how a stable final prompt is obtained.
>
> **Explanation of how the described problem is addressed:** Based on your suggestion, we provide more discussion on how the VR-DPGE and STORM strategy can reduce the variance of gradient estimation and on what the theorem indicates about the resulting prompts.
>
> **Discussion on experimental results:** In our response to you, we have added a discussion on the original experimental data and observations.
>
> We appreciate the time and effort you have invested in reviewing our work. Your comments have significantly strengthened our submission, and we hope that our responses address your concerns comprehensively.
>
> Thank you again for your thoughtful feedback and consideration.
>
> Best regards,
>
> The Authors

---

### Official Review · Reviewer_xsUR · 2024-11-01

**Soundness:** 3
**Presentation:** 3
**Contribution:** 2
**Rating:** 6
**Confidence:** 4

**Summary:**

This paper considers the problem of discrete prompt learning with black-box language models. The authors propose an unbiased variance-reduced doubly policy gradient estimator for optimizing prompts through policy gradient optimization. By incorporating the STORM variance reduction techniques, the algorithm obtains a stable prompt. Experimental results on several benchmark tasks show the effectiveness of the proposed algorithm.

**Strengths:**

1. This work studies the challenge of data imbalance in discrete prompt learning with black-box models, a significant issue in many practical applications.

2. Discrete prompt learning is widely necessary for practical applications with black-box models. The proposed method has the potential to serve as a fine-tuning approach for many commercial black-box models.

**Weaknesses:**

1. The proposed method, along with its theoretical analysis, largely combines existing techniques, which limits its technical novelty.

2. Several of the backbone models used in the experiments, such as RoBERTa-large and GPT2-XL from 2019, are outdated for current black-box applications. Additionally, the Llama 3 8B model is relatively small. In real-world applications, more commonly used models might include Llama 3 70B or GPT-4, accessed via API. How does the proposed algorithm perform on these state-of-the-art language models?

**Questions:**

see weakness 2

---

> ### Author Response · Authors · 2024-11-24
> **Response to Reviewer xsUR**
>
> We appreciate the time and effort put into reviewing our manuscript. Please find our responses to your concerns below:
>
> **weakness 1:**
> Thank you for your valuable feedback. Let us reiterate the technical novelty of our paper.
>
> 1. We acknowledge that our work is built upon the efforts of Zhangtong [1], who established a distribution optimization paradigm and introduced a policy gradient-based discrete prompt learning framework. We choose to follow it because it is one of the most influential frameworks in discrete prompt learning; however, we do not simply apply this framework in a straightforward manner. Despite the framework being a notable contribution, it also has significant shortcomings:
>
> * The explicit formulation for gradient estimator in [1] is flawed.
>
> * The policy gradient estimator used in the paper is biased, which introduces instability into the optimization process.
>
>   We resolve these significant shortcomings, which constitutes a key novelty of our work, as detailed below:
>
> * We correct the error in the explicit formulation for the gradient estimator in [1], with the detailed introduction provided in Appendix A of our paper
>
> * We first to provide unbiased gradient estimator leading to a more stable optimization process.
>
> 2. We do not simply combine existing techniques; instead, we carefully consider the suitability of various variance reduction techniques for imbalanced black-box prompt learning. Various variance reduction techniques, such as SVRG, require computing the full gradient at checkpoints using a giant batch of data, which is computationally expensive for black-box optimization and contradicts the simplicity and efficiency principles of prompt learning. Therefore, we ultimately adopt the STORM technique, which avoids the aforementioned issues and effectively reduces the dual variance.
>
> 3. Additionally, our paper is the first to address the challenge of prompt learning in imbalanced scenarios, which more closely reflect real-world applications. Classification tasks in such imbalanced settings hold significant practical importance.
> For instance, sensitive speech detection stands as a prime example of the profound and practical applications of classification tasks in real-world scenarios and inherently poses a natural data imbalance challenge [2].
> Furthermore, cutting-edge research [3] has turned its focus to utilizing LLMs for addressing deception detection, a task that inherently involves imbalanced classification. This research has culminated in the release of a series of novel high-quality datasets for the domain, highlighting the significance and industry-wide interest in advancing LLM-driven solutions for classification tasks under imbalanced data conditions.
>
> **weakness 2:**
> Thanks for your suggestion. In the revised version, we have included more experimental results to verify the effectiveness of mDP-DPG. Specifically, we add the BB (Burfoot and Baldwin dataset) [2], Job Scams and SMS datasets [3] to show performance of our method in real-world applications. In addition, we include comparisons with the recent baselines, APE [4] and EvoPrompt [5]. In this experiment, we use **GPT-4** to adapt our method to a true black-box model. The results are presented in the table below. For more details, please refer to the Appendix D.6 in the revision of our paper.
>
> | Length | Method | BB ($\tau=20$)| Job Scams ($\tau=20$)| SMS ($\tau=5$)|
> |:-:|:-:|:-:|:-:|:-:|
> | - | Manual Prompt  | .4333±.0191 | .5098±.0546     | .5252±.0131    |
> | $\leq$20 | APE  | .4968±.0398 | .5000±.0042     | .5268±.0184    |
> | $\leq$20 | EvoPrompt  | .5002±.0373 | .4972±.0064     | .5271±.0231    |
> | 5 | BDPL | .4486±.0146    | .4987±.0235     | .5200±.0114    |
> | 5 | mDP-DPG | **.5972±.1428**| **.5314±.0386** | **.5272±.0090**|
> | 20 | BDPL | .4403±.0244    | .5024±.0306     | .5135±.0064    |
> | 20 | mDP-DPG | **.5236±.1545**| **.5307±.0043** | **.5278±.0119**|
>
>
> [1] Diao et al., Black-box prompt learning for pre-trained language models.
>
> [2] Burfoot et al., Automatic Satire Detection: Are You Having a Laugh?
>
> [3] Boumber et al., Domain-agnostic adapter architecture for deception detection: Extensive evaluations with the difraud benchmark.
>
> [4] Zhou et al., Large language models are human-level prompt engineers.
>
> [5] Guo et al., Connecting large language models with evolutionary algorithms yields powerful prompt optimizers.

---

> ### Author Response · Authors · 2024-11-26
> **Did our response address your concern?**
>
> Dear reviewer, thank you again for your time and effort put into reviewing our manuscript. Please let us know if our responses have addressed your concerns. If you have remaining concerns, please don't hesitate to let us know and we are happy to address them. Thank you!

---

> ### Author Response · Authors · 2024-11-28
> **A kind follow-up**
>
> Dear Reviewer xsUR,
>
> Happy Thanksgiving!
>
> We greatly appreciate your insights, which have significantly improved our work. We hope that our responses and summary have adequately addressed your concerns.
>
> If you feel we've successfully addressed your points, we would be grateful for your support. Should you have any further questions, please don't hesitate to ask.
>
> Thank you again for your time and expertise in reviewing our work.

---

> ### Author Response · Authors · 2024-11-30
>
> Dear Reviewer xsUR,
>
> Thank you once again for your time and effort in reviewing our paper. As the discussion phase is nearing its end, we kindly request that you confirm whether our responses have addressed your concerns. We would appreciate it if you could fulfill your responsibilities as a reviewer by providing your feedback.
>
> Sincerely,
>
> The Authors

---

> > ### Comment · Reviewer_xsUR · 2024-12-01
> >
> > Thank you for your detailed response and clarifications. I appreciate the effort you put into addressing my concerns regarding the experiments on state-of-the-art large-scale models. I acknowledge that my initial concerns on this aspect have been address, and I am willing to raise my score to 6.
> >
> > However, I still find the technical contributions of the paper to be relatively minor. Specifically, compared to the work of Zhangtong et al. [1], the primary contribution of this paper appears limited. Both approaches employ variance-reduction techniques; while this paper adopts the STORM variance-reduction method proposed by Cutkosky and Orabona to enhance performance, the novelty of this substitution seems incremental.
> >
> > [1] Diao et al., Black-box prompt learning for pre-trained language models.
> > [2] Ashok Cutkosky and Francesco Orabona. Momentum-based variance reduction in non-convex sgd. Advances in neural information processing systems, 32, 2019.

---

> ### Author Response · Authors · 2024-12-01
> **Highlights of our contribution**
>
> Dear Reviewer xsUR,
>
> Thank you for your support of our article. We acknowledge that our work follows these two works. However, unlike these works, the main contributions of our work are as follows:
>
> - Our approach is the first to discuss how to learn discrete prompts for imbalanced data in LLM, which is a challenging task.
> - Our method provides a comprehensive theoretical analysis, whereas Zhongtong et al. in [1] did not offer any theoretical analysis results.
> - Following your suggestions, we conducted extensive experiments on the more realistically challenging GPT-4 model, and these experimental results demonstrate that our method has a performance advantage over BDPL [1].
>
> We once again thank you for the constructive review comments that have helped further improve the quality of our article and for your conscientious response.
>
> [1] Diao et al., Black-box prompt learning for pre-trained language models.

---

### Official Review · Reviewer_4JE3 · 2024-11-04

**Soundness:** 3
**Presentation:** 3
**Contribution:** 2
**Rating:** 6
**Confidence:** 3

**Summary:**

The authors propose a variance-reduced policy gradient descent method for black-box prompt learning. To deal with the class imbalance issue, the authors use AUC as the objective function. To accelerate the algorithm convergence, the authors propose to use two variance reduction techniques. The proposed algorithm has a convergence guarantee and good experimental performance.

**Strengths:**

1. To deal with the class imbalance issue, the authors propose to use AUC as the objective function.

2. To get fast convergence, the authors propose to reduce variance from both policy estimation and optimization.

**Weaknesses:**

1. The authors assume that the tokens in the prompt are sampled independently from their marginal distribution. However, it seems that tokens in the prompt should have some relation.

2. The proposed method can only for binary classification. How to extend to the multi-class? Do we need to train multiple different prompts?

**Questions:**

1. Can the authors describe why the probability they use assuming the tokens in the prompt sampled independently?

2. Is there a simple way to extend from binary classification to multi-class classification, since binary classification is rare in practice?

3. How concentrated the final solution P is? If the probability spread out to many different tokens, how can we generate a stable prompt to use?

---

> ### Author Response · Authors · 2024-11-24
> **Response to Reviewer 4JE3**
>
> We appreciate the time and effort put into reviewing our manuscript. Please find our responses to your concerns below:
>
> **weakness 1 & question 1:**
> Thank you for pointing out this concern. The assumption of independent sampling from the marginal distribution is a simplification that facilitates optimization in black-box settings, especially when the goal is to learn discrete prompts effectively. While this approach does not explicitly capture potential semantic relationships between tokens, we believe this assumption aligns with the broader objective of black-box optimization: improving downstream task performance without imposing additional constraints from token-level semantic relations. We acknowledge that this assumption may limit interpretability. Exploring techniques to model token relationships explicitly, while preserving the efficiency and simplicity of the current framework, is an exciting avenue for future work. Thank you for highlighting this potential direction.
>
> **weakness 2 & question 2:**
> Thanks for your question. Indeed, we have previously addressed the extension to multi-class tasks in the original paper, which can be found in Appendix D.1 of original paper (Appendix D.2 of the revision). Specifically, our method can be extended to multi-class datasets using micro-averaging. This involves calculating the AUC loss for each class against all others and average them over all classes.
>
> **question 3:**
> Really appreciate this question. In fact, each probability distribution in the final solution does not ideally converge to a delta distribution. In order to obtain a stable prompt, following the approach in [1], the token at a given position in the final prompt is selected as the token with the highest probability in its corresponding probability distribution.
>
> [1] Diao et al., Black-box prompt learning for pre-trained language models.

---

> > ### Comment · Reviewer_4JE3 · 2024-11-26
> >
> > Thank you for the explanation. I have no further concerns and updated my score accordingly.

---

> > > ### Author Response · Authors · 2024-12-03
> > >
> > > Thank you once again for dedicating your time and effort to reviewing our manuscript.

---

### Official Review · Reviewer_Ry7Y · 2024-11-04

**Soundness:** 3
**Presentation:** 3
**Contribution:** 2
**Rating:** 5
**Confidence:** 4

**Summary:**

This paper considers the (discrete) black-box prompt tuning for LLM for binary classification problems with imbalanced data. The main contributions are as follows:

1. The authors propose the mDP-DPG (momentum-based Discrete Prompt learning with Doubly Policy Gradient) approach. This approach combines pairwise AUC loss and variance reduction techniques to deal with imbalanced data.

2. A theoretical analysis is provided, showing that mDP-DPG achieves optimal convergence rates for imbalanced data scenarios, under some conditions such as bounded variance of the gradient.

3. Experiments on benchmark datasets (like CoLA and MRPC) and real-world imbalanced datasets (Amazon reviews) demonstrate that mDP-DPG outperforms baseline methods.

**Strengths:**

1. This is the first paper that studies how to address imbalanced data in NLP prompt learning.

2. The theoretical results might be insightful in the realm of black-box policy optimization.

**Weaknesses:**

1. Scenario

The paper considers prompt tuning involving:
- black-box LLM (e.g., API-only LLM)
- classification tasks
- imbalance data

However, prompt tuning for black-box LLMs is generally more relevant to complex generative tasks rather than simpler classification tasks. It is uncommon for entities to use models like GPT-4 primarily for text classification, especially when manually designed prompts can already yield strong performance on tasks like GLUE. This positioning may reduce the appeal of the proposed method to the main audience of ICLR.

2. Experiments

The paper does not include experiments on widely used API-only LLMs, such as GPT-3.5 or GPT-4. Additionally, it lacks comparisons with other existing black-box prompt tuning methods like APE [R1], EvoPrompt [R2], and ZOPO [R3].

[R1] Zhou, Y. et al. Large language models are human-level prompt engineers.

[R2] Guo, Q. et al. Connecting large language models with evolutionary algorithms yields powerful prompt optimizers.

[R3] Hu, W. et al. Localized Zeroth-Order Prompt Optimization.

**Questions:**

Please see "weakness".

---

> ### Author Response · Authors · 2024-11-24
> **Response to Reviewer Ry7Y**
>
> We appreciate the time and effort put into reviewing our manuscript. Please find our responses to your concerns below:
>
> **weakness 1:**
> Thank you for pointing out this important consideration regarding the scenario. We acknowledge that generative tasks have emerged as a pivotal focus in contemporary research. Nevertheless, classification tasks continue to hold paramount significance in the exploration of large language models, owing to their indispensable role in a wide array of practical applications, particularly in the context of more practical scenarios involving imbalanced data.
> Sensitive speech detection stands as a prime example of the profound and practical applications of classification tasks in real-world scenarios and inherently poses a natural data imbalance challenge [1].
> Furthermore, cutting-edge research [2] has turned its focus to utilizing LLMs for addressing deception detection, a task that inherently involves imbalanced classification. This research has culminated in the release of a series of novel high-quality datasets for the domain, highlighting the significance and industry-wide interest in advancing LLM-driven solutions for classification tasks under imbalanced data conditions.
> To demonstrate the superiority of our method and highlight its significance and performance in practical applications, we have conducted additional experiments on the aforementioned datasets. For detailed information, please refer to our response to weakness 2.
>
> **weakness 2:**
> Thanks for your suggestion. In the revised version, we have included more experimental results to verify the effectiveness of mDP-DPG. Specifically, we add the BB (Burfoot and Baldwin dataset) [1], Job Scams and SMS datasets [2] to show performance of our method in real-world applications. In addition, we include comparisons with the recent baselines, APE [3] and EvoPrompt [4]. As for ZOPO [5], since we can not access its source code, evaluating this method within the limited time available is challenging, and thus we have not included it in our comparisons for now. In this experiment, we use **GPT-4** to adapt our method to a true black-box model. The results are presented in the table below. For more details, please refer to the Appendix D.6 in the revision of our paper.
>
> | Length | Method | BB ($\tau=20$)| Job Scams ($\tau=20$)| SMS ($\tau=5$)|
> |:-:|:-:|:-:|:-:|:-:|
> | - | Manual Prompt  | .4333±.0191 | .5098±.0546     | .5252±.0131    |
> | $\leq$20 | APE  | .4968±.0398 | .5000±.0042     | .5268±.0184    |
> | $\leq$20 | EvoPrompt  | .5002±.0373 | .4972±.0064     | .5271±.0231    |
> | 5 | BDPL | .4486±.0146    | .4987±.0235     | .5200±.0114    |
> | 5 | mDP-DPG | **.5972±.1428**| **.5314±.0386** | **.5272±.0090**|
> | 20 | BDPL | .4403±.0244    | .5024±.0306     | .5135±.0064    |
> | 20 | mDP-DPG | **.5236±.1545**| **.5307±.0043** | **.5278±.0119**|
>
> [1] Burfoot et al., Automatic Satire Detection: Are You Having a Laugh?
>
> [2] Boumber et al., Domain-agnostic adapter architecture for deception detection: Extensive evaluations with the difraud benchmark.
>
> [3] Zhou et al., Large language models are human-level prompt engineers.
>
> [4] Guo et al., Connecting large language models with evolutionary algorithms yields powerful prompt optimizers.
>
> [5] Hu et al., Localized Zeroth-Order Prompt Optimization.

---

> ### Author Response · Authors · 2024-11-26
> **Did our response address your concern?**
>
> Dear reviewer, thank you again for your time and effort put into reviewing our manuscript. Please let us know if our responses have addressed your concerns. If you have remaining concerns, please don't hesitate to let us know and we are happy to address them. Thank you!

---

> ### Author Response · Authors · 2024-11-27
> **Response to Reviewer Ry7Y**
>
> Thank you again for your time and effort dedicated to reviewing our paper. To better address your concerns, let us reiterate the contribution of our paper.
>
> 1. Our work is built upon the efforts of Zhangtong [1], who established a distribution optimization paradigm and introduced a policy gradient-based discrete prompt learning framework. We choose to follow it because it is one of the most influential frameworks in discrete prompt learning; however, we do not simply apply this framework in a straightforward manner. Despite the framework being a notable contribution, it also has significant shortcomings:
>
> * The explicit formulation for gradient estimator in [1] is flawed.
>
> * The policy gradient estimator used in the paper is biased, which introduces instability into the optimization process.
>
> * [1] lacks a theoretical analysis of algorithmic convergence, indicating that the framework is not theoretically comprehensive.
>
> We resolve these significant shortcomings, which some of the key innovations in our work, as detailed below:
>
> * We correct the error in the explicit formulation for the gradient estimator in [1], with the detailed introduction provided in Appendix A of our paper
>
> * We first to provide unbiased gradient estimator leading to a more stable optimization process.
>
> * We have established a unique theoretical analysis framework for black-box discrete prompt learning with policy gradient, which makes the original framework's theoretical comprehensive.
>
> 2. We do not simply combine existing techniques; instead, we carefully consider the suitability of various variance reduction techniques for imbalanced black-box prompt learning. Various variance reduction techniques, such as SVRG, require computing the full gradient at checkpoints using a giant batch of data, which is computationally expensive for black-box optimization and contradicts the simplicity and efficiency principles of prompt learning. Therefore, we ultimately adopt the STORM technique, which avoids the aforementioned issues and effectively reduces the dual variance. The content of Lemma 4 demonstrates that STORM strategy effectively reduces the dual variance. The bound in Lemma 4 contains terms that decay with the momentum parameter $\theta_t$, showing that variance of the moving average $\boldsymbol{m}_{t+1}$ is systematically reduced over iterations, leading to a more stable and accurate approximation of the true gradient.
>
> 3. Additionally, our paper is the first to address the challenge of prompt learning in imbalanced scenarios, which more closely reflect real-world applications. Classification tasks in such imbalanced settings hold significant practical importance.
> For instance, sensitive speech detection stands as a prime example of the profound and practical applications of classification tasks in real-world scenarios and inherently poses a natural data imbalance challenge [2].
> Furthermore, cutting-edge research [3] has turned its focus to utilizing LLMs for addressing deception detection, a task that inherently involves imbalanced classification. This research has culminated in the release of a series of novel high-quality datasets for the domain, highlighting the significance and industry-wide interest in advancing LLM-driven solutions for classification tasks under imbalanced data conditions. Our latest supplementary experiments on **GPT-4** indicate that our method has advantages in handling the aforementioned tasks.
>
> [1] Diao et al., Black-box prompt learning for pre-trained language models.
>
> [2] Burfoot et al., Automatic Satire Detection: Are You Having a Laugh?
>
> [3] Boumber et al., Domain-agnostic adapter architecture for deception detection: Extensive evaluations with the difraud benchmark.

---

> ### Author Response · Authors · 2024-11-28
> **A kind follow-up**
>
> Dear Reviewer Ry7Y,
>
> Happy Thanksgiving!
>
> We greatly appreciate your insights, which have significantly improved our work. We hope that our responses and summary have adequately addressed your concerns.
>
> If you feel we've successfully addressed your points, we would be grateful for your support. Should you have any further questions, please don't hesitate to ask.
>
> Thank you again for your time and expertise in reviewing our work.

---

> ### Author Response · Authors · 2024-11-30
>
> Dear Reviewer Ry7Y,
>
> Thank you once again for your time and effort in reviewing our paper. As the discussion phase is nearing its end, we kindly request that you confirm whether our responses have addressed your concerns. We would appreciate it if you could fulfill your responsibilities as a reviewer by providing your feedback.
>
> Sincerely,
>
> The Authors

---

> ### Author Response · Authors · 2024-12-01
>
> Dear Reviewer Ry7Y,
>
> Thank you once again for dedicating your time and effort to reviewing our paper. The other reviewers have already affirmed the value of our work, and we believe we have sufficiently addressed your concerns. As the discussion phase is nearing its end, we hope you can further reconsider your evaluation of our manuscript.
>
> Sincerely,
>
> The Authors

---

> > ### Comment · Reviewer_Ry7Y · 2024-12-02
> >
> > Thank you for your detailed response and thorough explanation. I have raised the scores based on the added experiments and baselines, particularly for GPT-4.
> >
> > However, I remain unconvinced by the argument regarding the demand for classification tasks with imbalanced data. The authors cite one recent paper from 2024 and another from 2009. Upon reviewing these papers briefly, I respectfully disagree with the assertion that classification tasks, especially those involving imbalanced data, hold "paramount significance."
> >
> > Also, I believe LLM can handle complex classification tasks better than many traditional models, but I doubt that using **black-box LLM with training-based prompt tuning** is an appropriate choice, due to the consideration of security and the fast iteration of API-only LLMs (such as OpenAI GPT series).

---

> > > ### Author Response · Authors · 2024-12-03
> > >
> > > Dear Reviewer Ry7Y,
> > >
> > > We sincerely appreciate your willingness to engage in discussions with us and for your support of our paper. Thank you so much for acknowledging our additional results and your remaining questions.
> > >
> > > First, we emphasize that classification tasks hold paramount significance due to their critical applications in real-world scenarios, e.g., fake news detection.
> > > It is well known that news often exerts a significant influence on public behavior and opinions, and it can even impact major political events such as the U.S. elections. For example, during the heated stage of the 2016 U.S. presidential election, social media was flooded with information that was difficult to distinguish as true or false. Young people from a small town in Macedonia even made a fortune by creating fake news about the U.S. election [1]. Although Trump ultimately won the election in 2016, according to the Washington Post's fact-checking agency, 70% of his statements during the campaign were completely untrue "misformations" or plausible "disinformation" [2]. By classifying and filtering out fake news in advance, such mechanisms play a crucial role in safeguarding national stability and promoting societal development. Even former U.S. President Obama publicly condemned fake news for threatening the democratic process in the United States [3]. Therefore, we believe that using LLMs for classification is both meaningful and valuable. In the latest additional comparative experiments conducted during the rebuttal period, we also included experimental results on imbalanced fake news datasets BB and Job Scams. Our results show a clear advantage compared to the baselines, outperforming the second-best results by nearly 10% and 3%, respectively.
> > >
> > > For security, black box prompt tuning allows users to store private data labels and customize prompts locally, thereby preventing potential data leakage and safeguarding users' commercial interests [4]. For the fast iteration of API-only LLMs, black box prompt tuning is a lightweight approach to adapting LLMs for downstream tasks, making the cost of prompt re-training caused by API updates entirely manageable. Additionally, we have included comparative experimental results on GPT-4 against BDPL, which can be considered as a version of mDP-DPG that does not account for imbalanced data.
> > >
> > > | Length | Method | Book ($\tau=10$) | Elec ($\tau=10$) | BB ($\tau=20$)| Job Scams ($\tau=20$)| SMS ($\tau=5$)|
> > > |:-:|:-:|:-:|:-:|:-:|:-:|:-:|
> > > | 20 | BDPL | .5058±.1096| .5018±.1115 | .4403±.0244 | .5024±.0306 | .5135±.0064 |
> > > | 20 | mDP-DPG | **.5063±.0459** | **.5105±.0095** | **.5236±.1545**| **.5307±.0043** | **.5278±.0119**|
> > >
> > > Sincerely,
> > >
> > > The Authors
> > >
> > > [1] Samanth	Subramanian,	Inside	the	macedonian	fake	news	complex,	https://www.wired.com/2017/02/veles-macedonia-fake-news/
> > >
> > > [2] Jonathan	Freedland,	Post-truth	politicians	such	as	Donald	Trump
> > > and	Boris	Johnson	are	no	joke, https://www.theguardian.com/
> > > commentisfree/2016/may/13/boris-johnson-donald-trump-post-
> > > truth-politician
> > >
> > > [3] Melissa De Witte, Taylor Kubota, and Ker Than,‘Regulation has to be part of the answer’to combating online disinformation, Barack Obama said at Stanford event, https://news.stanford.edu/stories/2022/04/disinformation-weakening-democracy-barack-obama-said
> > >
> > > [4] Diao et al., Black-box prompt learning for pre-trained language models

---

### Author Response · Authors · 2024-11-28
**Summary**

Dear Reviewers,

Happy Thanksgiving!

We appreciate your thorough evaluations and constructive feedback on our submission. We have carefully addressed all your comments and provided responses to each of them:

**Importance of the scenario:** We discuss the practical significance of classification tasks on imbalanced data and provide the example of recent advancement in the research community. For further details, please refer to our response to reviewer Ry7Y.

**Additional experiments:** Based on the suggestion of reviewer Ry7Y and xsUR, we add experiments on real-world tasks using the **GPT-4** and compare our method with recent related works. Please refer to the Appendix D.6 in the revision for details.

**Reiteration of contributions:** To aid in better understanding of the paper, we have reiterated the contributions, which are threefold: 1. Theoretical and technical improvements to address the limitations of the original framework. 2. A proposed approach for reducing dual variance. 3. A practical perspective (imbalanced data) on the application of prompt learning. For further details, please refer to our responses to reviewer Ry7Y and xsUR.

**Justification of the assumption:** To address the concerns of reviewer 4JE3 and pkzH, we explain the justification of the assumption that the token distributions are independent in our response to them.

**Extension to multi-class problem:** We have previously addressed the extension to multi-class tasks in the original paper, which can be found in Appendix D.1 of original paper (Appendix D.2 of the revision).

**Clarification of distribution convergence:** To address the concerns of reviewer 4JE3 and pkzH, we clarify the distribution convergence effect in the experiments and explain how a stable final prompt is obtained.

**Explanation of how the described problem is addressed:** Based on reviewer pkzH's suggestion, we provide more discussion on how the VR-DPGE and STORM strategy can reduce the variance of gradient estimation and on what the theorem indicates about the resulting prompts.

**Discussion on experimental results:** In our response to reviewer pkzH, we have added a discussion on the original experimental data and observations.

We appreciate the time and effort you have invested in reviewing our work. Your comments have significantly strengthened our submission, and we hope that our responses address your concerns comprehensively.

Thank you again for your thoughtful feedback and consideration.

Best regards,

The Authors

---

### Meta-Review · Area_Chair_fN7x · 2024-12-18

**Metareview:**

This paper proposes a prompt learning method for classification problems using LLMs, which specializes in handling the class imbalance issue. The method is based on the previous approach which adopted policy gradient for prompt learning, and incorporates variance reduction and momentum to achieve improved performance.

The reviewers in general agree that the studied problem (i.e., class imbalance in LLM-based classification problems) is novel, and theoretical guarantees are nice and insightful contributions.

On the other hand, the reviewers also pointed out some important concerns. A notable concern is that using LLMs for classification tasks is a less common paradigm compared to using them in generation tasks, which I also agree with. I understand that text classification is important in some scenarios (such as fake news detection), however, there are plenty of existing works which adopt specialized approaches for text classification, so using LLMs for this purpose may not be a very appealing choice. This may limit the practical significance of the proposed method. Another concern is the assumption of token independence, which is pointed out by multiple reviewers and also acknowledged by the authors to be a limitation. The lack of technical novelty of the proposed method is also brought up.

As a result, the reviewers did not champion the paper, and hence rejection is recommended.

**Additional Comments On Reviewer Discussion:**

During rebuttal, the reviewers engaged in discussions with the authors regarding some important concerns about the paper, such as the significance of the studied problem of using LLM for classification. After these discussions, some important concerns remained (see above).

---

### Decision · Program_Chairs · 2025-01-22

Reject